# Identification of novel human microcephaly-linked protein *Mtss2* that mediates cortical progenitor cell division and corticogenesis through *Nedd9-RhoA*

Aurelie Carabalona[1]*, Henna Kallo[2], Maryanne Gonzalez[2], Liliia Andriichuk[2], Ellinoora Elomaa[3,4], Florence Molinari[5], Christiana Fragkou[1], Pekka Lappalainen[6], Marja W Wessels[7], Juha Saarikangas[2,3,4], Claudio Rivera[1,2]*

[1]Aix Marseille Univ, INSERM, INMED, Marseille, France; [2]Neuroscience Center, HiLIFE, University of Helsinki, Helsinki, Finland; [3]Helsinki Institute of Life Science, HiLIFE, Helsinki, Finland; [4]Faculty of Biological and Environmental Sciences, University of Helsinki, Helsinki, Finland; [5]Aix Marseille Univ, Inserm, MMG, Marseille, France; [6]Institute of Biotechnology, HiLIFE, University of Helsinki, Helsinki, Finland; [7]Department of Clinical Genetics, Erasmus MC, University Medical Center Rotterdam, Rotterdam, Netherlands

**\*For correspondence:**
aurelie.carabalona@inserm.fr (AC);
claudio.rivera@helsinki.fi (CR)

**Competing interest:** The authors declare that no competing interests exist.

## eLife Assessment

This **important** contribution to the field evaluated the function of the cytoskeletal protein ABBA in mediating key aspects of mitosis of neuronal precursor cells. The authors provide **compelling** evidence that ABBA interactions with its signaling partners is related to the development of at least some cases of microcephaly-a developmental anomaly associated with intellectual disability and other neurological findings.

**Abstract** The cerebral cortex, which is responsible for higher cognitive functions, relies on the coordinated asymmetric division cycles of polarized radial glial progenitor cells for proper development. Defects in the mitotic process of neuronal stem cells have been linked to the underlying causes of microcephaly; however, the exact mechanisms involved are not fully understood. In this study, we present a new discovery regarding the role of the membrane-deforming cytoskeletal regulator protein called Mtss2 (also known as MTSS1L/ABBA) in cortical development. When Mtss2 was absent in the developing brain, it led to a halt in radial glial cell proliferation, disorganized radial fibers, and abnormal migration of neuronal progenitors. During cell division, Mtss2 localized to the cleavage furrow, where it recruited the scaffolding protein Nedd9 and positively influenced the activity of RhoA, a crucial regulator of cell division. Notably, we identified a variant of *Mtss2* (R671W) in a patient with microcephaly and intellectual disability, further highlighting its significance. The introduction of this mutant Mtss2 protein in mice resulted in phenotypic similarities to the effects of *Mtss2* knockdown. Overall, these findings offer valuable mechanistic insights into the development of microcephaly and the cerebral cortex by identifying *Mtss2* as a novel regulator involved in ensuring the accurate progression of mitosis in neuronal progenitor cells.

## Introduction

Development of the central nervous system occurs by proliferation of neural progenitor cells, migration, and differentiation of their progeny over substantial distance. These different steps require extensive membrane remodeling and cytoskeleton dynamics (*Ayala et al., 2007*). Neuronal migration involves the coordinated extension and adhesion of the leading process along the radial glial scaffold with the forward translocation of the nucleus, which requires regulation of centrosome and microtubule dynamics by different proteins (*Ayala et al., 2007*; *Higginbotham and Gleeson, 2007*; *Lambert de Rouvroit and Goffinet, 2001*). However, little is known about the molecular mechanisms underlying membrane dynamics during neuronal migration and morphogenesis.

Cell membrane curvature is a micromorphological change involved in many important cellular processes, including endocytosis, exocytosis, and migration (*Liu et al., 2015*). Recent studies demonstrated that members of an extended protein family, characterized by the presence of a membrane binding and deforming Bin-Amphiphysin-Rvs (BAR) domain function at the interface between the actin cytoskeleton and plasma membrane during the formation of membrane protrusions or invaginations (*Doherty and McMahon, 2008*; *Frost et al., 2009*; *Takano et al., 2008*). These proteins have been shown to be important for proper neuronal morphogenesis (*Qualmann et al., 2011*) and can either generate positive membrane curvature to facilitate the formation of plasma membrane invaginations (e.g. BAR, N-BAR, and F-BAR domain proteins), or induce negative membrane curvature to promote the formation of plasma membrane protrusions (I-BAR and IF-BAR domain proteins) (*Guerrier et al., 2009*; *Mattila et al., 2007*). An important part of the membrane-deforming mechanism includes the interaction with phosphatidylinositol-4,5-bisphosphate (PI(4,5)P2)-rich membranes through their BAR domain and also with actin cytoskeleton through Wasp Homology-2 domain (WH2).

Regulation of the actin cytoskeleton plays a crucial role in several developmentally important cellular processes, including neuronal migration, axonal and dendritic extension and guidance, and dendritic spine formation (*Spillane et al., 2011*). We previously discovered that MIM (Missing-In-Metastasis; Mtss1), a member of the I-BAR protein family, has an important role in bending the dendritic plasma membrane in order to form dendritic spines and sculpt their morphology (*Saarikangas et al., 2015*). In a more recent study, a closely related protein Mtss2 (Mtss1L) was found to be important for exercise-induced increase in synaptic potentiation in the hippocampus (*Chatzi et al., 2019*). Although an important role of MIM and Mtss2 in synapse formation and plasticity starts to emerge, information about the role of the early expression of this protein family during development is poor.

Mutations in BAR domain proteins are increasingly found in mental retardation disorders. For instance, altered expression and mutation of F-BAR family proteins, SrGAP2 and -3, are known to be associated with neurological disorders (mental retardation, the 3p syndrome, schizophrenia, early infantile epileptic encephalopathy, and severe psychomotor delay; *Wilson et al., 2011*; *Carlson et al., 2011*; *Endris et al., 2002*; *Waltereit et al., 2012*; *Saitsu et al., 2012*). Furthermore, previous whole-exome sequencing of prescreened multiplex consanguineous families identified *MTSS2* as a likely candidate linked to neurogenic disorders (*Alazami et al., 2015*), suggesting that some BAR-containing proteins might play important functions during brain development. Despite Mtss2's potential role in neuronal development, very little is known about how this protein may lead to neurological pathology.

Regulation of the actin cytoskeleton and plasma membrane dynamics in glial cells of the developing brain has remained poorly understood. Interestingly, the I-BAR protein Mtss2 is the only glia-enriched regulator that has been reported so far (*Saarikangas et al., 2008*).

We sought to determine the consequences of altered expression of *Mtss2* in radial glial progenitor (RGP) cells on cell cycle progression and neurogenesis and to test for subsequent neuronal migration. To address these issues, we used in utero electroporation to silence *Mtss2* with short hairpin RNA (*shRNA*) in embryonic mouse brains. Downregulation of Mtss2's expression had a striking effect on RGP cell progression, resulting in a blocking of cytokinesis and a subsequent apoptosis. Biochemical analysis demonstrated that the cytokinesis block was likely due to an absence of interaction with the adhesion docking protein named Nedd9 (aka CAS-L or HEF1) that is well known to be a positive regulator of RhoA signaling in mitosis (*Dadke et al., 2006*). Using FRET-based monitoring of RhoA activity, we also found a significant role of Mtss2 on RhoA activity. Moreover, we observed a defect in neuronal migration associated with an increase in neuronal apoptosis. Importantly, we identified one patient harboring a missense variant in *MTSS2* sequence and showing specific features such as small head circumference, mental retardation, autism spectrum disorder (ASD), and craniofacial dysmorphism.

We provide evidence that the (2011C>T(R671W)) patient mutation displays similar phenotypes and decreased Mtss2 expression.

Overall, our data highlights the critical role of *Mtss2* in proper cortical development in neurogenesis and migration and provides evidence that *MTSS2* is a new causative gene for cortical malformation associated with ASD.

## Materials and methods

### Plasmids construction

We utilized a single *shRNA* that targeted the coding sequence for Mtss220. It is worth noting that the rat and mouse Mtss2 sequences share a high degree of similarity, with a 95% homology. The specific region targeted by the *shRNA* displayed 100% homology. Both Mtss2 and Nedd9 *shRNA* were designed using the siRNA Wizard Software (InvivoGen). As a negative control, we employed a nontargeting *shRNA*.

These *shRNA*s were subcloned into an mU6pro vector (*Yu et al., 2002*). Additionally, a pCAG-RFP vector was co-injected to facilitate the visualization of fluorescent cells. Mtss2 mouse full length with silent mutations (*Saarikangas et al., 2008*), MTSS2 human full length, and human MTSS2-R671W carrying vectors were made by GeneCust (France) and subcloned into the pCAGIG-IRES-GFP vector (Addgene, Cambridge, MA, USA).

### Bioinformatic methods

MTSS2 coding sequences were obtained from NCBI. The multiple sequence comparison was done by Log-Expectation (MUSCLE) method and visualized with DNASTAR MegAlign Pro. The impact of MTSS2 R671W on the structure and function of a human MTSS2 protein was predicted using Poly-Phen-2 and HumVar model (*Adzhubei et al., 2010*).

### Human subjects

Families were identified through our clinical and research programs, personal communication, as well as the MatchMaker Exchange (MME), including GeneMatcher (*Sobreira et al., 2015*) (http://www.genematcher.org). Informed consent for publication and analysis of photos, imaging, and clinical data was obtained from the patients' legal guardians. Brain magnetic resonance imaging (MRI) studies were performed on the subject and reviewed by the investigators in accordance with the Medical Ethics Committee of Erasmus Medical Center, Rotterdam, the Netherlands (Department Ethical permission number: MEC-2012387) to conduct human exome studies.

### RhoA activity assay

Fluorescence from Raichu-Rac1 was imaged using a confocal microscope (LSM 710/Axio; Carl Zeiss) controlled by ZEN 2011 software (Carl Zeiss) and equipped with water immersion objective (×63/1.0 NA; Carl Zeiss). Transfected cells were illuminated with an argon laser (Lasos; Lasertechnik GmbH) at 458 nm (CFP and FRET) and 514 nm (YFP) using 0.5–10% of full laser power. Emission was collected at 461–519 nm (CFP) and 519–621 nm (FRET and YFP). The pinhole was fully opened. Scanning was performed in XY mode using 6× digital zoom that resulted in a pixel XY size of 68×68 nm$^2$. FRET signal was calculated as reported elsewhere (*Adzhubei et al., 2010*). In brief, using ImageJ 1.48v software, images were background-subtracted, and FRET image generated using three images: MDonor (CFP excitation and CFP emission filters), MIndirectAcceptor (CFP excitation and YFP emission filters), and MDirectAcceptor (YFP excitation and YFP emission filters), using the following equation: FRET = (MIndirectAcceptor − MDonor × β − MDirectAcceptor × (γ − αβ))/(1 − βδ). Coefficients α, β, δ, and γ were obtained by independent control experiments analyzing cells expressing CFP or YFP as described previously (*van Rheenen et al., 2004*). Coefficients α, δ, and γ were determined in cells expressing only YFP (acceptor) and calculated according to equations: α=MDonor/MDirectAcceptor; γ=MIndirectAcceptor/MDirectAcceptor; δ=MDonor/MIndirectAcceptor. Coefficient β was determined in cells expressing only CFP (donor) and calculated according to the equation: β=MIndirectAcceptor/MDonor. After the generation of the FRET image, the intensity of the FRET signal was calculated using a mask obtained from a corresponding YFP image and covering neuronal cell bodies. The images were also analyzed using Pix-Fret (*Feige et al., 2005*).

## In utero electroporation

Animal experiments were performed in agreement with European directive 2010/63/UE and received approval N°: APAFIS#2797 from the French Ministry for Research and in agreement with the National Animal Experiment Board, Finland (license number ESAVI/18276/2018). Plasmids were transfected using intraventricular injection followed by in utero electroporation (*Saito and Nakatsuji, 2001*; *Tabata and Nakajima, 2001*). Timed pregnant C57BL/6N mice (Janvier Labs; E14; E0 was defined as the day of confirmation of sperm-positive vaginal plug) received buprenorphine (Buprecare, 0.03 mg/kg) and were anesthetized with sevoflurane/isoflurane (4–4.5% induction, 2% anesthesia maintenance) 30 min later. For pain management, bupivacaine (2 mg/kg) was administered via a subcutaneous injection at the site of the future incision. Uterine horns were exposed, the lateral ventricle of each embryo was injected using pulled glass capillaries and a microinjector (Picospritzer II; General Valve Corporation, Fairfield, NJ, USA) with Fast Green (2 mg/mL; Sigma, St Louis, MO, USA) combined with the different DNA constructs at 1.5 µg/µL or with 1.5 µg/µL *shRNA* constructs together with 0.5 µg/µL pCAGGS-RFP and Anilline-GFP48. Plasmids were further electroporated by discharging a 4000 mF capacitor charged to 50 V with a BTX ECM 830 electroporator (BTX Harvard Apparatus, Holliston, MA, USA). The voltage was discharged in five electrical pulses at 950 ms intervals via 5 mm tweezer-type electrodes laterally pinching the head of each embryo through the uterus. The mice were followed up postsurgically and given buprenorphine (Buprecare 0.03 mg/kg) if needed. The number of animals has been calculated on the basis of the requirement for adequate numbers of brain slices and sections for sufficient imaging to provide statistically significant data on the effects of RNAi, small molecules, and other reagents used in the proposed analysis, plus controls.

## Organotypic cultures and live imaging

To assess the impact of altered MTSS2 expression in the RGP cells on neurogenesis and cell cycle progression, we utilized a combination of in utero electroporation and time-lapse imaging of organotypic cultures. The organotypic cultures were prepared as described (*Tabata and Nakajima, 2001*) with the following exceptions. After surrounding the brains (E17) with 4.5% UltraPure Low Melting Point Agarose (Invitrogen, 16520-100), 350-µm-thick slices were cut with Vibratome 1000 Plus Sectioning System (No. 054018). Before placing the slices on a coated (83 µg/mL Laminin, Sigma, L2020; 8.3 µg/mL Poly-L-Lysine, Sigma, P4707) cell culture insert (Millicell, PICMORG50), sections were carefully cleaned from agarose to ensure adequate oxygenation. The inserts with slice cultures were transferred into a six-well plate containing 1 ml of warmed slice culture medium. For live imaging, we used the Andor Dragonfly 550 high-speed Spinning Disc confocal equipped with Nikon Perfect Focus System, motorized XY stage, and environmental chamber. The system uses Fusion 2.0 software (version 2.1.0.48). The image acquisition was executed using Andor iXon 888 U3 EMCCD camera, ×20 magnification with voxel size $0.65×0.65×0.9524\ \mu m^3$ (channel/excitation: red fluorescent protein [RFP]/561 nm, GFP/488 nm). Time-lapse imaging of organotypic cultures was started at E17+DIV0 with intervals of 10 min and total duration of 15 hr (*Wiegreffe et al., 2017*). The data was collected from 6 pups (2 scramble, 4 *shRNA*), 17 slices (5 scramble, 12 *shRNA*), and 24 areas (9 scramble, 15 *shRNA*; some slices were recorded from two different areas). For quantification of Anillin-GFP (*Hesse et al., 2012*) positive cells, Z-projection images of individual time points were made and positive cells tracked over time using the plugin TrackMate in ImageJ (*Tinevez et al., 2017*) (NIH, Bethesda, MD, USA). All experiments were conducted in accordance with the French and Finnish ethical committee which approved all procedures (No APAFiS#38335 and ESAVI/3183/2022, respectively). All animal experiments complied with the ARRIVE guidelines and were carried out in accordance with the U.K. Animals (Scientific Procedures) Act, 1986 and associated guidelines, EU Directive 2010/63/EU for animal experiments. All methods were performed following the relevant guidelines and regulations.

## Immunostaining of brain slices

Mice brains were fixed (E17) in Antigen Fix (Diapath). Brain slices were sectioned coronally (80 µm) on a vibratome (Leica microsystems). Brain slices were washed with PBS (phosphate-buffered saline pH 7.4) and stained in PBS 0.3% Triton X-100 supplemented with 5% of donkey serum. Primary antibodies were incubated overnight at 4°C, sections were then washed with PBS and incubated in secondary antibodies for 2 hr at room temperature. Antibodies used in this study were: Mtss2 (homemade),

Vimentin (Millipore, MAB3400), Ki67 (Millipore, AB9260), phospho-histone 3 (PH3) (Abcam, ab14955), Pax6 (BioLegend, PRB-278P).

## Cell lines, culture, and transfection

We used the C6 rat glioma cell line in this study. The cells were obtained from ATCC (Cat# CCL-107). Cell line identity was authenticated by short tandem repeat profiling. Mycoplasma contamination was routinely assessed using PCR-based assays, and all cultures used in this study tested negative.

The C6 cell line is included on the International Cell Line Authentication Committee (ICLAC) list of commonly misidentified cell lines. Nevertheless, its continued use is scientifically justified due to its extensively characterized properties, relevance in modeling glial and glioma-like cellular functions, and its widespread use in neurobiological research. Experimental outcomes were interpreted with full awareness of this context.

Rat C6 glioma cells were cultured at 37°C under a humidified atmosphere with 5% $CO_2$ in complete medium (DMEM supplemented with 10% fetal bovine serum [Sigma] and 100 units/mL antibiotics/anti-mycotics). Cells were transfected using the Neon Transfection System (Life Technologies) according to the manufacturer's protocol. Briefly, cells were trypsinized and counted using the SCEPTER cell counter (Millipore). Electroporation was performed using 500,000 cells and a total of 5 μg of DNA, consisting of a reporter plasmid encoding RFP (1:5 ratio) and *shRNA*s or mismatch control constructs. The electroporation settings were 1860 V, 1 pulse, 20 ms. Transfected cells were subsequently cultured in six-well plates for 2 days prior to RNA extraction.

## Immunocytochemistry

HEK293T cells were fixed in Antigen Fix (Diapath) for 15 min and blocked at room temperature for 1 hr with 5% normal goat serum, 0.3% Triton X-100 in PBS, and incubated overnight at 4°C with a set of antibodies against Mtss2 (homemade), Phalloidin (Thermo Fisher, A12379), Nedd9 (Abcam, ab18056), Hoechst (Thermo Fisher, 62249). Cells were then washed with PBS and incubated in secondary antibodies for 2 hr at room temperature. The definition of cortical regions was guided by stainings against Cux1 (Santa Cruz, sc-13024) and Ctip2 (Abcam, ab18465).

## RT-PCR and qPCR

Total RNA was isolated from C6 using RNeasy Plus Mini kit, and cDNA was synthesized using the Quantitect Reverse Transcription kit, according to the manufacturer's protocol (QIAGEN). Quantitative PCR (qPCR) was performed on a LightCycler 480 using SYBR-Green chemistry (Roche) and specific primers for Mtss2 (F: CGAGACTCGCTGCAGTATTCC and R: CCATTCACAGAGTAGCAGTC G, 113 bp), Nedd9 (QIAGEN, QT01610385) and Cyclophilin A (QIAGEN, QT00177394) as a control probe. qPCR was performed with 5 μL of diluted cDNA template, specific primers (0.6 μM), and SYBR Green I Master Mix (7.5 μL) at a final volume of 15 μL. Each reaction was performed at an annealing temperature of 60°C and for 50 cycles. Reactions were performed in duplicate, and melting curve analysis was performed to assess the specificity of each amplification. A standard curve was performed for each gene with a control cDNA diluted at different concentrations. Relative expression was assessed with the calculated concentration with respect to the standard. All experiments were performed in triplicate.

## Microscopy and image analysis

All images were collected with an IX80 laser scanning confocal microscope (LSM 710/Axio; Carl Zeiss) controlled by ZEN 2011 software (Carl Zeiss). Cells and brain sections were imaged using a ×60 1.42 N.A. oil objective or a ×10 0.40 N.A. air objective. All images were analyzed using ImageJ (NIH, Bethesda, MD, USA). To quantify the total number of C6 cells alive and in mitosis on 12 mm diameter coverslips 2 and 3 days after Mtss2 *shRNA* and control transfections, we have arbitrarily designed four identical regions comprised of nine squares using ZEN software acquisition system and counted manually the total number of alive and mitotic cells in each condition. All cell quantifications in brain slices were initially normalized to electroporation efficacy. For analysis of morphological changes in vimentin staining, images were analyzed using the directionality plugin in ImageJ, and the values corresponding to dispersion extracted.

## Statistical data

Statistical analyses were performed with Prism (GraphPad Software, La Jolla, CA, USA). A two-sample Student's t-test was used to compare means of two independent groups if the distribution of the data was normal. If the values come from a Gaussian distribution (D'Agostino-Pearson omnibus normality test), the parametric unpaired t-test with Welch's correction was used. But when the normality test failed, the non-parametric Mann-Whitney test was used. Significance was accepted at the level of $p < 0.05$.

No statistical methods were used to predetermine sample sizes, but our sample sizes are similar to those generally employed in the field. No randomization was used to collect all the data, but they were quantified blindly.

## Y2H screen

Yeast two-hybrid (Y2H) screening was performed by Hybrigenics Services (Paris, France). The Mtss2 coding sequence containing amino acids (aa) 1–715 was cloned into pB27 (N-LexA-bait-C fusion) and pB66 (N-GAL4-bait-C fusion) and sequence verified. These constructs were used as baits to screen the mouse embryo brain RP2 prey library. A total of 91 million (pB27) and 31 (pB66) million interactions were analyzed, and the detected interactions were assigned with a statistical confidence score, the Predicted Biological Score (PredBioScr) (https://www.hybrigenics-services.com/).

## Co-immunoprecipitation

C6 cells or cortical tissue were washed with ice-cold PBS and lysed using lysis buffer (20 mM Tris-HCl [pH 8.0], 137 mM NaCl, 10% glycerol, 1% Triton X-100, 2 mM EDTA) with 50 µg/mL PMSF (Roche) and a protease inhibitor cocktail (Roche). Cell lysate was agitated for 30 min at 4°C and centrifuged at 12,000 rpm for 20 min at 4°C. The supernatant was collected and protein concentrations were determined with a Pierce BCA Protein Assay kit, according to the manufacturer's protocol (Thermo Scientific).

Before co-IP, lysates were precleared with washed Pierce Protein A/G Magnetic Beads (Thermo Scientific) overnight in gentle shaking at 4°C, to remove nonspecific binding. Beads used for co-IP were washed and pre-blocked with 3% BSA in PBS overnight in gentle shaking at 4°C. Washing steps were done using a wash buffer (1% BSA, 150 mM NaCl, 0.1% Tween-20 in TBS).

For co-IP, 1000 µg of protein from precleared lysate was incubated with 20 µL of specific primary antibodies (Mtss2: homemade; Nedd9: Abcam, ab18056) overnight in gentle shaking at 4°C. Negative controls were incubated without primary antibodies under the same conditions. Immunocomplexes were pulled down by incubating the antigen-antibody mixture with pre-blocked beads for 1 hr in gentle shaking at room temperature. Beads were washed three times with a wash buffer and once with purified water before elution. Bound proteins were eluted from the beads by incubating them in 1x Laemmli buffer containing 150 mM NaCl for 10 min at room temperature.

Original C6 cell lysate and immunoprecipitated proteins were resolved by SDS-PAGE (4–20% Mini-PROTEAN TGX Gel; Bio-Rad) and transferred using Trans-Blot Turbo Mini PVDF Transfer Packs and Transfer System (Bio-Rad). Membranes were blocked with TBS containing 0.1% Tween-20 and 5% milk and blotted with primary antibodies overnight in shaking at 4°C. HRP-conjugated secondary antibodies used in this study were Rabbit anti-mouse IgG and Goat anti-rabbit IgG (Invitrogen). Protein bands were visualized using the ChemiDoc MP imaging system (Bio-Rad).

## Flow cytometry

To measure cell cycle length, $4 \times 10^5$ cells were seeded in six-well plates, transfected, and cultured for 24 hr. Plasmid transfection was performed with jetPEI by following the Batch HTS protocol suggested in the data sheet of the product. Briefly, transfection complexes, which consist of both DNA and jetPEI reagent diluted in NaCl, are mixed with the suspension of cells obtained after trypsinization and distributed in the wells for culture over 24 hr. Then, cell cycle synchronization was induced by 24 hr serum starvation and restarted using fetal calf serum supplemented DMEM. The cells were detached with trypsin+EDTA and fixed with 3.7% formaldehyde after 24 hr in culture. $5 \times 10^5$ fixed cells from each stop point were stained in 200 µL of Hoechst 33342 (1:2000 dilution) for 30 min, washed with PBS 1X, and diluted in a flow buffer. Acquisition of 100,000 total events was performed in NovoCyte

Quanteon 4025, and the analysis was performed in FlowJo v10.8 or NovoExpress 1.6.1 software. Graphs and statistical analysis were performed in GraphPad Prism 5.

## Results

### *Mtss2* knockdown disrupted radial glia morphology

Previous studies of the expression pattern of *Mtss2* in the developing brain have disclosed a conspicuous expression in radial glia. In order to investigate the function of *Mtss2* in the developing brain, we used a mouse in utero RNAi approach to knock down *Mtss2* at E14. For this purpose, we opted for *shRNA* knockdown using a previously validated targeting sequence (*Saarikangas et al., 2008*). Revalidation of the knockdown efficiency of this target with three different Mtss2-shRNA (Mtss2-*shRNA*1-3) in rat C6 glioma cells showed a 70–80% reduction of Mtss2 mRNA and protein levels with Mtss2-*shRNA*3 (*Figure 1—figure supplement 1*).

To determine whether knockdown of Mtss2 expression alters neuronal migration, Mtss2-*shRNA*3 combined with an RFP construct was introduced into neural progenitor cells of mouse neocortex by in utero electroporation at E14 and evaluated at E16–18 (*Figure 1A*). Electroporation of E14 targets the progenitors that will migrate and populate cortical layers II/III neurons. In E17 brain sections, we observed that some neurons previously electroporated 3 days earlier with the scrambled construct reached the cortical plate as expected (*Figure 1B*). However, in utero expression of Mtss2-*shRNA*3 induced a significant arrest of cells within the ventricular zone (VZ) (*Figure 1A–C*; scramble: 24.3±6.4%; n=9, Mtss2-*shRNA*3: 63.7±2.9%; p<0.0001; n=8). Surprisingly, we also observed a striking decrease in the total number of RFP-positive cells in brain sections electroporated with Mtss2-*shRNA*3 at E17 and E18 (*Figure 1D–E*; scramble: 969.2±109.0 cells/mm$^2$; n=9, Mtss2-*shRNA*3: 311.3±28.61 cells/mm$^2$; p<0.0001; n=8). These results indicate an important role of Mtss2 in the proliferation and migration of cortical neuronal progenitors.

We also evaluated the impact of *Mtss2* knockdown on radial glial morphology by imaging vimentin-stained *Mtss2* knockdown cells at 3 days post-electroporation. All E17 *Mtss2* knockdown brains, but not in age-matched controls, displayed disruption of radial glial fibers in the VZ/subventricular zone (SVZ) with increased dispersion in Mtss2 knockdown (*Figure 1D and F*; *Figure 1—figure supplement 2*; scramble 14.27 ± 6% and Mtss2-*shRNA*3 40.8±19, n=20; p=0.0078). Control vimentin-positive cells displayed the characteristic features and polarized morphology of radial glia with processes extending from the VZ to the pial surface. In contrast, in *Mtss2* knockdown brains, vimentin-positive cells exhibited misoriented apical processes, detached end feet in the ventricular surface (*Figure 1D*) and decreased number of radial glial progenitor cells in the VZ (*Figure 1F*; scramble: 126.5±13,4 cells, n=8; Mtss2-*shRNA*3: 65.1±26.2 cells, n=7; p=0.0033). All together, these data show that the loss of Mtss2 can disrupt radial glial cell morphology, thus affecting their progenitor function.

These data suggest that *Mtss2* is required for radial glial organization, neuronal migration, regulation of neuronal proliferation, and cell survival. Importantly, these results are in accord with the reported association of *MTSS2* with developmental neurological diseases, including intellectual disability and microcephaly (*Alazami et al., 2015*).

### MTSS2 is required for cell cycle progression of radial glia progenitors

Development of the cerebral cortex occurs through a series of stages, beginning with RGPs. These stem cells exhibit an unusual form of cell-cycle-dependent nuclear oscillation between the apical and the basal regions of the VZ, known as interkinetic nuclear migration (INM). RGP cells undergo mitosis only when they reach the SVZ (*Kosodo, 2012*; *Spear and Erickson, 2012*). It has been shown that inhibition of INM disrupts RGP cell cycle progression (*Tsai et al., 2010*; *Hu et al., 2013*; *Carabalona et al., 2016*).

To test the effect of altered radial glial cell morphology on cell cycle progression, we have quantified the nuclear distance from the VS for RGP cells after electroporation of scrambled or Mtss2 RNAi (*Figure 2A*). We observed an accumulation of Mtss2-*shRNA*3 RGP cells close to the ventricular surface (0–10 mm; scramble: 30.2 ± 3.2%, n=7; Mtss2-*shRNA*3: 38.3 ± 2.3%, n=7; p=0.000156), which can correspond to mitotic cells, but also further away (>30 mm; scramble 12.0±3.3%, n=7; Mtss2-*shRNA*3: 21.9±2.4%, n=7; p=0.000023), which support our previous observations on cells detaching from the VS (*Figure 1D and F*). Furthermore, we observed that Mtss2-*shRNA*3 decreased

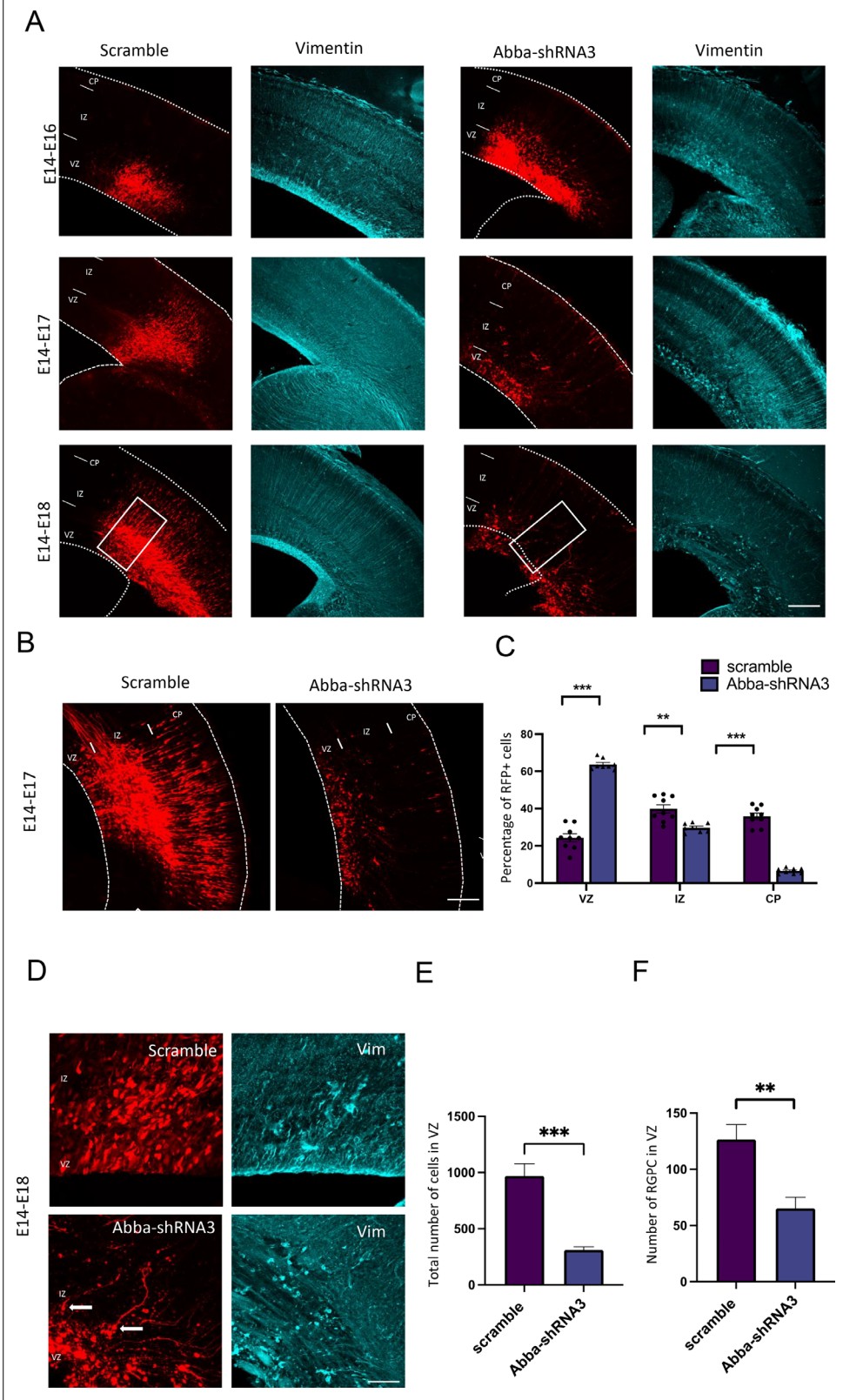

**Figure 1.** In utero knockdown of *Mtss2* expression is linked to radial glial disruption and alters neuronal migration. (**A**) Representative coronal sections of E16–18 mice brains electroporated at E14 with either scramble or Mtss2-*shRNA*3 and immunostained for vimentin (in blue). In Mtss2-*shRNA*3 brain sections, vimentin staining revealed a marked disruption of radial glial apical fibers. (**B**) Representative neocortical coronal sections showing migration

*Figure 1 continued on next page*

*Figure 1 continued*

of transfected cells 3 days after electroporation at E14 with either scramble or Mtss2-*shRNA*. (**C**) Quantification of red fluorescent protein (RFP)-positive cell distribution in the cortex at E17 (VZ/SVZ: scramble 24.33 ± 2.13%, Mtss2-*shRNA*3 63.71 ± 1.03%; IZ: scramble 39.80 ± 2.22%, Mtss2-*shRNA*3 29.71 ± 0.85%; CP: scramble 35.86 ± 1.73%; Mtss2-*shRNA*3 6.58 ± 0.53%; n=9 scramble and 8 Mtss2-*shRNA*3; VZ/SVZ: p=<0.0001: IZ: p=0.0002; CP: p=<0.0001) showing a significant increase of Mtss2 knockdown cells in SVZ/lower IZ. (**D**) Higher magnifications of regions indicated by a square in . (**E**) Total number of RFP-positive cells in mouse neocortex at E17 (scramble 969.2 ± 109.0%; Mtss2-*shRNA*3 311.3 ± 28.61%; n=9 scramble and 8 Mtss2-*shRNA*3) showing a striking reduction of RFP-positive cells in Mtss2 knockdown electroporated brains. Error bars represent mean ± s.d. **p<0.002, ***p<0.001. (**F**) Quantification of radial glial progenitor cells at the VZ (scramble 126.50 ± 13.36%, Mtss2-*shRNA*3 65.14 ± 9.91%). Mean ± s.d. VZ: ventricular zone, SVZ: subventricular zone, IZ: intermediate zone, CP: cortical plate. Error bars represent mean ± s.d. **p<0.002. Scale bars: 100 µm (**A,B**), 50 µm (**D**).

The online version of this article includes the following figure supplement(s) for figure 1:

**Figure supplement 1.** Alteration of Mtss2 expression by RNAi.

**Figure supplement 2.** Distribution and directionality of radial glial progenitor (RGP).

the percentage of the RGP cells marker Pax6 (*Figure 2B and C*; scramble: 31.9±0.6%, n=8; Mtss2-*shRNA*3: 21.1±0.6%, n=5; p=<0.0001). We have then tested for cell cycle effects and found a decrease in the percentage of Mtss2 *shRNA*3-expressing RGP cells positive for the cell cycle marker Ki67 (*Figure 2D and E*; scramble: 46.5±0.8%, n=7; Mtss2-*shRNA*3: 23.7±1.369%, n=5; p=<0.0001) and also for the late G2/M phase marker PH3 (*Figure 2F and G*; scramble: 4.5±0.1764%, n=9; Mtss2-*shRNA*3: 1.264±0.5191%, n=5; p=<0.0001). More detailed examination of the impact of Mtss2-*shRNA*3 on cell cycle progression using flow cytometry showed an accumulation in S-phase. Interestingly, these effects were rescued by overexpression of Mtss2-FL (*Figure 2—figure supplement 1A*; G1: scramble: 45.2±1.4%, Mtss2-*shRNA*3: 37.4±1.0%, Mtss2-*shRNA*3+Mtss2-FL: 42.8±1.3%; S: scramble: 34.3±2.8%, Mtss2-*shRNA*3: 48.2±2.7%, Mtss2-*shRNA*3+Mtss2-FL: 38.9±3.4; G2: scramble: 17.1±2.1%, Mtss2-*shRNA*3: 14.4±2.7%, Mtss2-*shRNA*3+Mtss2-FL: 15.9±1.9%; n=10; p=<0.001). These results indicate that Mtss2 plays a major role in coordinating cell cycle progression in radial glia.

## Mtss2 localizes to the cleavage furrow and is important for cytokinesis

To further characterize the role of Mtss2 in mitotic progression, we synchronized C6 rat glioma cell line and stained with Mtss2 antibody. We found that Mtss2 localized to the putative apical membrane initiation site and cytokinetic bridge, proximal to the contractile actin ring during late telophase and cytokinesis (*Figure 3A and B*, respectively). The actomyosin contractile ring is formed at the plasma membrane in response to accumulation of PI(4,5)P2, to generate constricting force to separate the two daughter cells (*Miller, 2011*). Mtss2 is known to interact with PI(4,5)P2 via its membrane deforming I-BAR domain (*Saarikangas et al., 2008*) that has the capacity to deform and recognize the membrane curvature.

To determine the impact of the loss of Mtss2 on cytokinesis, we quantified the number of C6 cells in cytokinesis 3 days after transfection with Mtss2-*shRNA*3 or scramble *shRNA*. Importantly, we found a two-and-a-half-fold increase in the percentage of Mtss2-*shRNA*3-expressing cells in cytokinesis (*Figure 3B and D*; scramble 0.8285±0.1161%, n=12; Mtss2-*shRNA*3 2.078±0.2682%, n=12; p=0.0358), suggesting a prominent role of Mtss2 in cytokinesis. To validate this in vivo, we performed live imaging in cortical organotypic slices from animals that were in utero electroporated with either Mtss2-*shRNA*3 or scramble vectors. Analysis of live-imaging recording showed a strong mitotic block at the ventricular surface (*Figure 3C*; *Figure 3—video 1*; *Figure 3—video 2*), as well as significant accumulation of a marker of mitotic progression Anillin-GFP in RFP cells during the recording period (*Figure 3—figure supplement 1*, scramble: 100.2 ± 15% and Mtss2-*shRNA*3: 152.6 ± 22, n=20; p=0.036). We then examined the impact of cytokinesis block on cell survival and found a lower total number of cells 4 days after transfection with Mtss2-*shRNA*3 compared with control scramble (*Figure 3E*; scramble: 380.3±48.16%, n=12; Mtss2-*shRNA*3: 155.8±22.32%, n=12; p=<0.0001). Taken together, these data provide evidence that Mtss2 is required for the completion of mitosis, and its absence results in a reduced number of cell progeny.

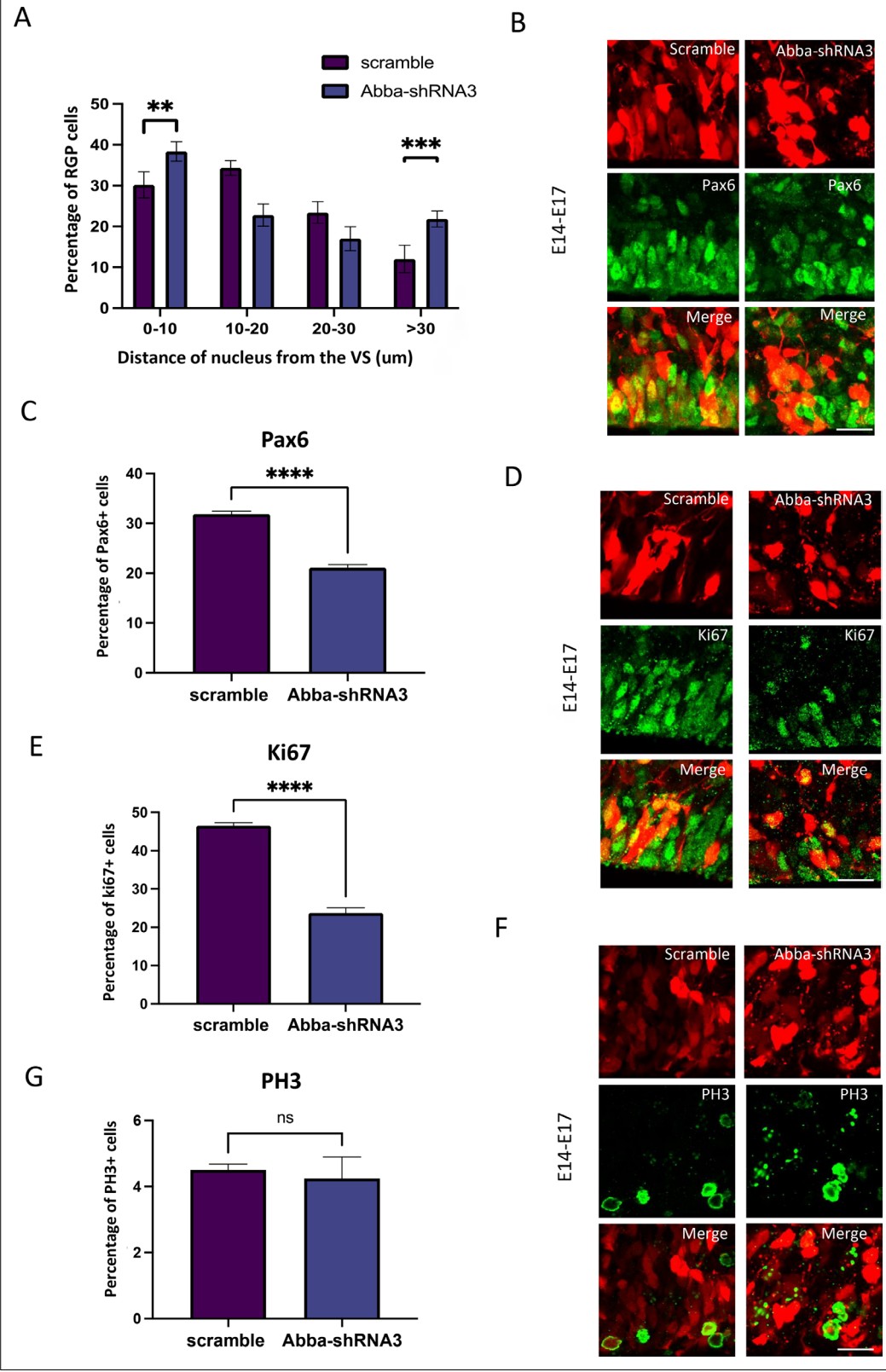

**Figure 2.** Mtss2 downregulation-induced inhibition of basal nuclear migration in radial glial progenitor (RGP) cells blocks cell cycle progression. (**A**) Quantification of the nuclear distance from the ventricular surface (VS) for RGP cells after electroporation of scrambled or Mtss2-*shRNA*3. Nuclear distribution of red fluorescent protein (RFP) RGP cells was significantly altered after electroporation of Mtss2-*shRNA*3 (0–10: scramble, 30.20 ± 1.21%;

*Figure 2 continued on next page*

*Figure 2 continued*

Mtss2-*shRNA*3, 38.34 ± 0.89%; 10–20: scramble, 34.35±0.69%; Mtss2-*shRNA*3, 22.80 ± 1.03%; 20–30: scramble, 23.42 ± 1.00%; Mtss2-*shRNA*3, 17.00 ± 1.10%; >30: scramble, 12.04 ± 1.27%; Mtss2-*shRNA*3, 21.85 ± 0.74%; 0–10: p=0.000156; 10–20: p=<0.000001; 20–30: p=0.001030; >30: p=0.000023). (**B–C**) E14 mice brains were subjected to in utero electroporation with either scramble or Mtss2-*shRNA*3. Brains were then fixed at E17 and stained with the RGP cell marker Pax6. We observe an important decrease of Pax6 RGP cells expressing Mtss2-*shRNA*3 (scramble, 31.87 ± 0.57%; Mtss2-*shRNA*3, 21.08 ± 0.63%; n=8 scramble; 5 Mtss2-*shRNA*3). (**D–G**) E14 mice brains were subjected to in utero electroporation with either scramble or Mtss2-*shRNA*3. Brains were then fixed at E17 and stained with the cell cycle marker Ki67 (**D**) and phospho-histone 3 (PH3) (**F**). We observe a striking decrease in the percent of cycling (**E**: Ki67; scramble, 46.54 ± 0.79%, n=7; Mtss2-*shRNA*3, 23.71 ± 1.37%, n=5 Mtss2-*shRNA*3) and but not in mitotic (**G**:PH3; scramble, 4.50 ± 0.17%; Mtss2-*shRNA*3, 4.25 ± 0.65%) cells with low expression of Mtss2. . Error bars represent mean ± s.d. **p<0.002, ***p<0.001. Scale bar represents 20 μm.

The online version of this article includes the following figure supplement(s) for figure 2:

**Figure supplement 1.** Flow cytometry analysis of C6 cells.

## Mtss2 recruits Nedd9 to the cytokinetic bridge and is required for RhoA activation

In earlier work, we have shown that MTSS2 is mainly expressed through E10.5–12.5 in the floorplate structure formed by radial glia (*Saarikangas et al., 2008*). To investigate the mechanism by which Mtss2 operates in mitosis, we performed a Y2H screen in E10.5–12.5 mouse brain embryo library using full-length mouse Mtss2 as a bait. The resulting analysis identified three high (PredBioScr: B/C) confidence interactors: the E3 ubiquitin ligase Beta-Trcp2 (PredBioScr: B), scaffolding protein Nedd9 (PredBioScr: C), the transcription factor Otx2 (PredBioScr: C) (*Figure 4A*; *Supplementary file 1*). Among these candidate proteins, Nedd9 (aka HEF-1 or Cas-L) caught our interest as it was previously reported in breast cancer cells to localize to the cleavage furrow during cytokinesis where it activates RhoA, as well as abnormal expression of Nedd9 results in cytokinesis defect (*Dadke et al., 2006*; *Zhang and Wu, 2015*; *Zhong et al., 2012*; *Figure 4A-B*).

We first tested whether Mtss2 and Nedd9 physically interact, as suggested by the Y2H screen. To this end, we performed immunoprecipitation experiments from embryonic day 18 cortical tissue, as well as C6 cells, using anti-Nedd9, as well as anti-Mtss2 antibodies. The precipitated Nedd9-Mtss2 complex was separated on SDS-PAGE, followed by western blot with anti-Mtss2 and anti-Nedd9 anti-bodies, respectively. Indeed, Mtss2 precipitated with Nedd9 in both directions, but not using control beads from brain cortex homogenates (*Figure 4C*) and C6 cells (*Figure 4—figure supplement 1*). Nedd9 stability is not affected by the availability of Mtss2, as western blots show no changes in Nedd9 expression levels in Mtss2-*shRNA*3-expressing cells (data not shown). Interestingly, however, when we analyzed the localization of Nedd9, we found that Mtss2 was required for targeting Nedd9 to the cleavage furrow/cytokinetic bridge in C6 cells 72 hr post-transfection (*Figure 4D*). These data suggest that Mtss2 recruits Nedd9 to the cleavage furrow during RGP cell division, which fits to its role in sensing curved PI(4,5)P2-rich membranes. Nedd9 plays a role in RGP cytokinesis in vivo (*Dadke et al., 2006*; *Zhang and Wu, 2015*; *Zhong et al., 2012*). To knock down Nedd9, we used *shRNA* targeting Nedd9, which reduced mRNA levels by 75% in C6 cells (*Figure 4—figure supplement 1A*). We used a previously characterized Nedd9-*shRNA* to determine whether the knockdown of Nedd9 expression alters RGP cell morphology and function. To this end, Nedd9-*shRNA* combined with an RFP construct was introduced into neural progenitor cells of mouse neocortex by in utero electroporation at E14 and stained with the radial glial cell marker, vimentin. Interestingly, similarly to Mtss2 knockdown, Nedd9 downregulation resulted in disruption of INM as a larger portion of progenitors accumulated close to the ventricular surface (*Figure 4E and F*; scramble: 25 ± 2.30% n=8, Nedd9-*shRNA*: 46.92 ± 2.78%; n=7; p=<0.0001). Using the late G2/M phases marker, PH3, we observed similar results as with Mtss2 *shRNA* and point mutation and no difference in number of PH3-positive cells (*Figure 4—figure supplement 1C*; scramble: 4.50 ± 0.18%, n=9; Nedd9-*shRNA*: 5.07 ± 0.35%, n=6, p=0.1135). The role of Nedd9 in cell cycle progression is consistent with previous published results (*Dadke et al., 2006*; *Zhang and Wu, 2015*; *Zhong et al., 2012*) and provides evidence that both Mtss2 and Nedd9 function to ensure correct mitotic completion of RGP cells.

Finally, to determine whether Mtss2 is important for Nedd9 activity and how Mtss2 and Nedd9 converge to regulate the cell cycle, we assessed the activity of RhoA, a downstream signaling molecule

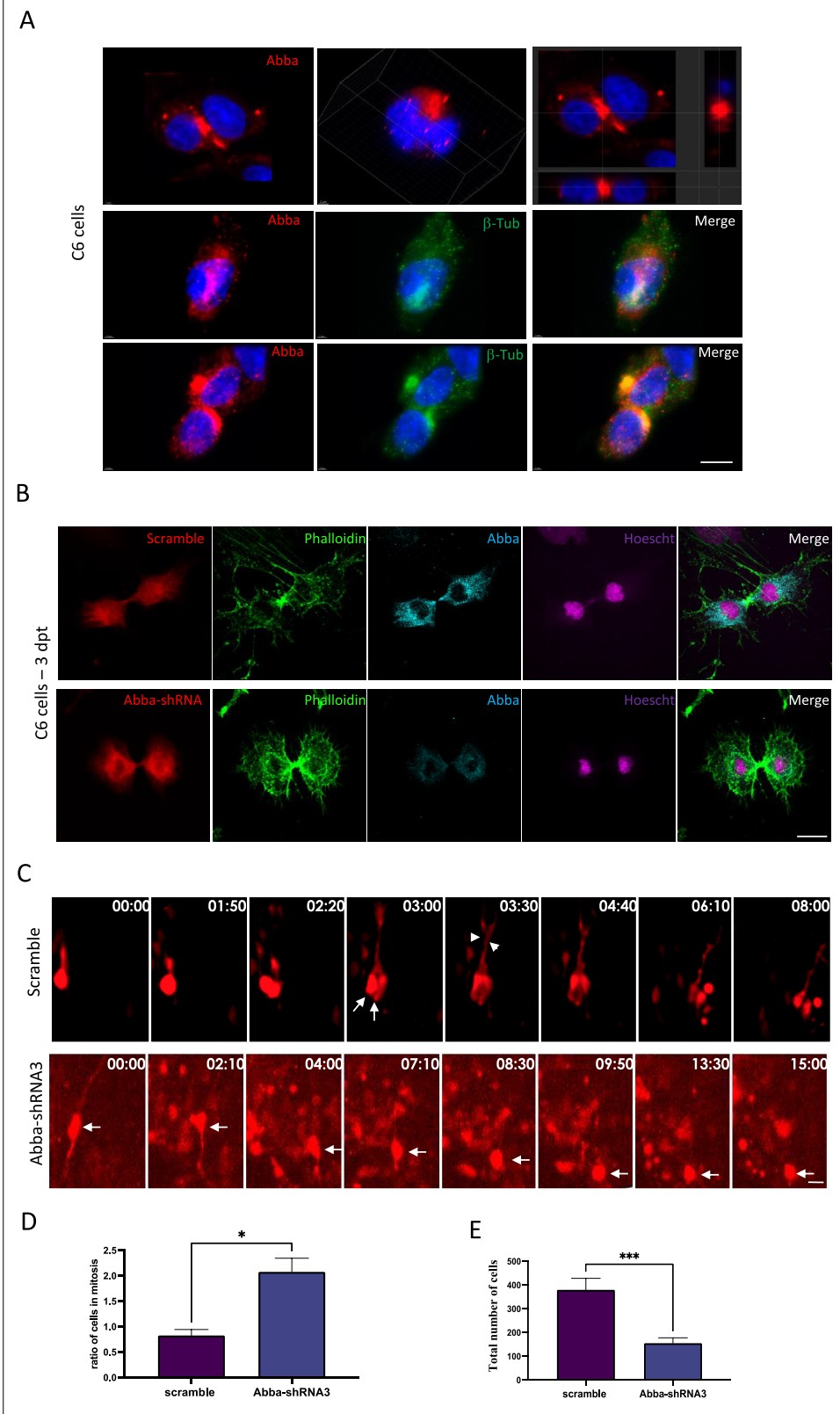

**Figure 3.** Mtss2 downregulation blocks cytokinesis. (**A**) Cellular distribution of Mtss2 in C6 cells during cell division. Upper panel shows two example immunofluorescent microscopy images of Mtss2 (red). Left-most figure shows the projection images at the cross line of the image. Mid and lower panels show the distribution of Mtss2 at different cell division stages in relation to β-tubulin (green) aggregation. (**B**) Immunofluorescence microscopy images from

*Figure 3 continued on next page*

*Figure 3 continued*

C6 cells 72 hr after transfection with Mtss2-*shRNA*3 show an absence of Mtss2 expression at cytokinesis. (**C**) Brain slices prepared from in utero electroporated animals with either Mtss2-*shRNA* or scramble were cultured at E17 and subjected to live imaging for a duration of 15–20 hr. The length of the time lapse was adjusted as necessary to capture significant events during interkinetic nuclear migration (INM). In the case of the scramble group, a radial glial progenitor (RGP) cell underwent a single mitotic event at the ventricular surface of the brain slice, observed at time points 1:50 and 2:20. At time point 3:00, two cells can be observed (indicated by arrows), along with the presence of basal fibers (indicated by arrowheads). Conversely, in the Mtss2 knockdown group, the nucleus of the RGP (indicated by an arrow) initially underwent apical interkinetic nuclear migration (INM) and subsequently divided at 8:30. Remarkably, the resulting daughter cells remained at the ventricular surface for at least 7 hr, suggesting that the absence of Mtss2 hinders cells from exiting the mitotic state. (**D**) Mtss2-*shRNA*3-expressing cells show an increase in cytokinesis (scramble 0.8285±0.1161%, n=12; Mtss2-*shRNA*3 2.078±0.2682%, n=12). (**E**) Quantification of the total number of red fluorescent protein (RFP)-positive cells in the neocortex illustrating the impact of cytokinesis block on cell survival. We observed a lower total number of cells 4 days after transfection with Mtss2-*shRNA*3 compared with control scramble (scramble 380.3±48.16%; n=12, Mtss2-*shRNA*3 155.8±22.32%; p=<0.0001; n=12). Error bars represent mean ± s.d. *p<0.03, **p<0.002, ***p<0.001. Scale bars: 20 μm.

The online version of this article includes the following video and figure supplement(s) for figure 3:

**Figure supplement 1.** Quantification of Anillin expression changes during 15 hr live-imaging recordings.

**Figure 3—video 1.** Symmetric division.

https://elifesciences.org/articles/92748/figures#fig3video1

**Figure 3—video 2.** Absence of division in Mtss2-depleted progenitor cells.

https://elifesciences.org/articles/92748/figures#fig3video2

---

of Nedd9 (*Singh et al., 2007*), which is important for cytokinetic actomyosin ring assembly and dynamics (*Singh et al., 2007*; *Jarin et al., 2019*; *Glotzer, 2001*). To determine RhoA activity, we used a GFP-based FRET construct for RhoA designated 'Ras and interacting protein chimeric unit' Raichu-RhoA. In order to test the efficacy of the assay, we used Calpeptin, a RhoA activator which, as expected, increased RhoA activity (% change compared to Raichu-RhoA: Calpeptin 132.2±3.2; n=33; p=0.002), confirming the functionality of our assay (*Figure 5*). Importantly, when Mtss2 was silenced by *shRNA*, there was a significant 31.5% decrease in Raichu-RhoA activity as compared to the control *shRNA*-treated cells (*Figure 5B*; scramble: 103.1.03±5.3; n=102, Mtss2-*shRNA*3: 68.5±2.3; n=46; p<0.0001). In addition, this effect was rescued by the co-expression of *shRNA*-insensitive plasmid carrying the full-length sequence of Mtss2 (Mtss2-*shRNA*+Mtss2-FL 139.8±19.6; n=26; p<0.0001 compared to Mtss2-*shRNA*3). Mtss2-FL did not change RhoA activity significantly compared to scramble-expressing cells (Mtss2-FL 105.2±16.0; n=37). Similar results were found when quantifying Raichu-RhoA activity in regions corresponding to the cleavage furrow (*Figure 5C*; scramble: 76.3±27.6; n=18, Mtss2-*shRNA*3: 44.8±17.2; n=15, Mtss2-*shRNA*+MTSS2-FL: 102.5±49.7 n=23, Mtss2-FL: 81.6±387.0; n=22; p<0.0001, Mtss2-R671W: 135.7±62.4; n=15). These data provide support for the function of Mtss2 in spatial assembly of cytokinetic signaling through the Nedd9-RhoA axis.

## Identification of *MTSS2* missense variant R671W in an individual with neurodevelopmental syndrome

Mutations in genes affecting mitotic progression of neuronal progenitors have been linked to neurodevelopmental disorders (*Klingler et al., 2021*). Importantly, we identified a patient carrying a novel heterozygous missense variant in *MTSS2* coding *MTSS1l/MTSS2* gene. This mutation (2011C>T(R671W)) affects residues that are conserved from zebrafish to human and reside in exon 15 (*Figure 6A and B*). Consistent with a de novo origin of the variant, it was not found in the parents of the affected individual or in normal controls. This patient presents some characteristic craniofacial dysmorphism as mild deep-set eyes, thickened helices, synophrys, narrow forehead, bitemporal narrowing, small bilateral epicanthal folds, upslanting palpebral fissures, medial flaring of eyebrows, flat midface, V-shaped palate, and dysmorphic ears. Brain MRI sequences showed the presence of a microcephalic brain associated with a smaller head linked to mild IQ. The patient also suffered from attention-deficit/hyperactivity disorder (ADHD) (*Figure 6C*, *Supplementary file 2*). Moreover, recent publications found six additional patients carrying the same *MTSS2* variant that shared similar brain malformation (*Huang et al., 2022*; *Corona-Rivera et al., 2023*). This variant replaces an arginine with

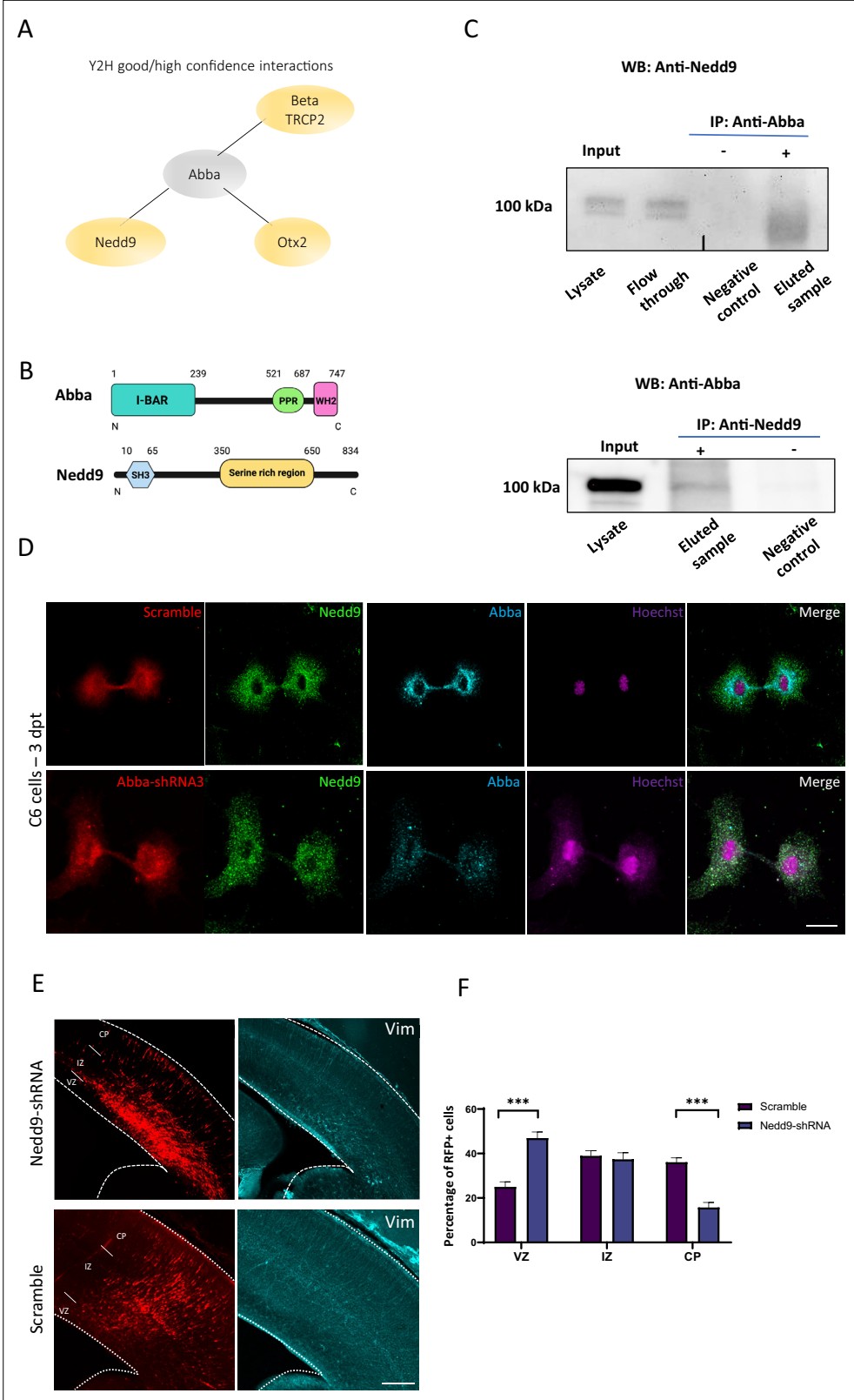

**Figure 4.** Role of Mtss2-Nedd9 complex in radial glial progenitor (RGP) cells division. (**A**) Good and high confidence interactions detected in yeast two-hybrid screen performed in mouse brain embryo library using full-length mouse Mtss2 as a bait. (**B**) Schematic representation of the Mtss2 and Nedd9 proteins. (**C**) E18 cortical homogenates were subjected to a direct pull-down assay with endogenous Mtss2 and Nedd9. Nedd9

*Figure 4 continued on next page*

*Figure 4 continued*

co-immunoprecipitated with Mtss2, but not in control pull-downs showing the specificity of the interaction. (**D**) Immunocytochemistry on C6 cells 72 hr after transfection with Mtss2-*shRNA*3 shows that lack of Mtss2 did not decrease Nedd9 expression (green) but rather its localization at the cleavage furrow/cytokinetic bridge stage. Scale bars: 20 μm. (**E**) Representative coronal sections of E17 mice brains transfected at E14 with Nedd9-*shRNA* and immunostained for vimentin (in blue). Vimentin staining revealed a marked disruption of radial glial apical fibers. (**F**) Quantification of red fluorescent protein (RFP)-positive cell distribution in the cortex at E17 (ventricular zone/subventricular zone [VZ/SVZ]: scramble 25 ± 2.30%, Nedd9-*shRNA* 46.92 ± 2.78%; IZ: scramble 38.94 ± 2.32%, Nedd9-*shRNA* 37.37 ± 2.96%; CP: scramble 36.06 ± 1.95%; Nedd9-*shRNA* 15.71 ± 2.25%; n=8 scramble and 7 Nedd9-*shRNA*; VZ/SVZ: p=<0.0001: IZ: p=0.0002; CP: p=<0.0001) showing a significant increase of Nedd9 knockdown cells in SVZ/lower IZ. Scale bars: 100 μm (**E**), 50 μm (**D**).

The online version of this article includes the following source data and figure supplement(s) for figure 4:

**Source data 1.** Western blot source data.

**Source data 2.** Nedd9 WB source data.

**Source data 3.** Abba WB source data.

**Figure supplement 1.** Nedd9 *shRNA* efficiency test in C6 cells and effect on phospho-histone 3 (PH3) positive cell in vivo.

**Figure supplement 1—source data 1.** Abba WB source data.

---

tryptophan that has a large aromatic group that could lead to destabilization of a tertiary structure. Indeed, bioinformatic prediction using AlphaFold predicts (pLDDT) with high 84.4 confidence score that R671 is located in an alpha helical fold close to the C-terminal end of the protein (*Field et al., 2005*). Furthermore, by using Poly-Phen-2 structure and sequence-based algorithm (*Adzhubei et al., 2010*) to predict the impact of R671W substitution on the structure and function of MTSS2, we found a high probability that the mutation is damaging to the protein function (naïve Bayes probability score 0.952) (*Figure 6D*).

## Expression of R671W human variant results in defects in mitosis and neuronal migration

To test the effects of the human missense variant in mouse brain development, we introduced the human *MTSS2* cDNA encoding either wild-type or patient mutation R671W into developing mouse brains by in utero electroporation at E14 followed by imaging and quantification of targeted cells at E17. Interestingly, whereas wild-type MTSS2 (MTSS2-FL) expression showed no defects in radial glial morphology or neuronal migration, the R671W mutant expression resulted in disorganized radial glial fibers and accumulation of multipolar neurons in the SVZ/lower IZ with very few bipolar neurons (*Figure 6E*). When analyzing the distribution of electroporated neuron progenitors, we observed clear accumulation in the VZ/SVZ (*Figure 6F*; MTSS2-FL 18.67 ± 1.44%; n=8, MTSS2-R671W 39.77 ± 2.73% n=6; p=0.000; *Figure 2—figure supplement 1B and C*). We also found a significant increase in RGP nuclei at the VS (*Figure 6G* ; MTSS2-FL 32.56 ± 2.58%; n=8, MTSS2-R671W 50.04 ± 2.04%; n=6; p=0.000001; *Figure 2—figure supplement 1B and C*) and changes in the distance of the nucleus from the VZ. Interestingly, no difference in accumulation of PH3+ cells was found (Mtss2-human-WT 4.50 ± 0.17%; n=9, Mtss2-human-mutation 4.71 ± 1.98%; p=0.76; n=5), indicative of a dysfunction in neuronal maturation but a less prominent impact on cell division. In line with these results, the effect of MTSS2-R671W on RhoA activation was not significantly different from scramble-expressing cells (142±17.9; n=15). The data presented indicate that the abnormal expression of the R671W mutation, rather than the normal protein, leads to abnormal development of the developing cortex. These findings offer evidence suggesting a potential connection between the missense mutation in MTSS2 discovered in the patient and abnormal brain development. However, it is unlikely that the mechanism behind this phenotype is a result of MTSS2-R671W acting in a dominant-negative manner on cell cycle progression.

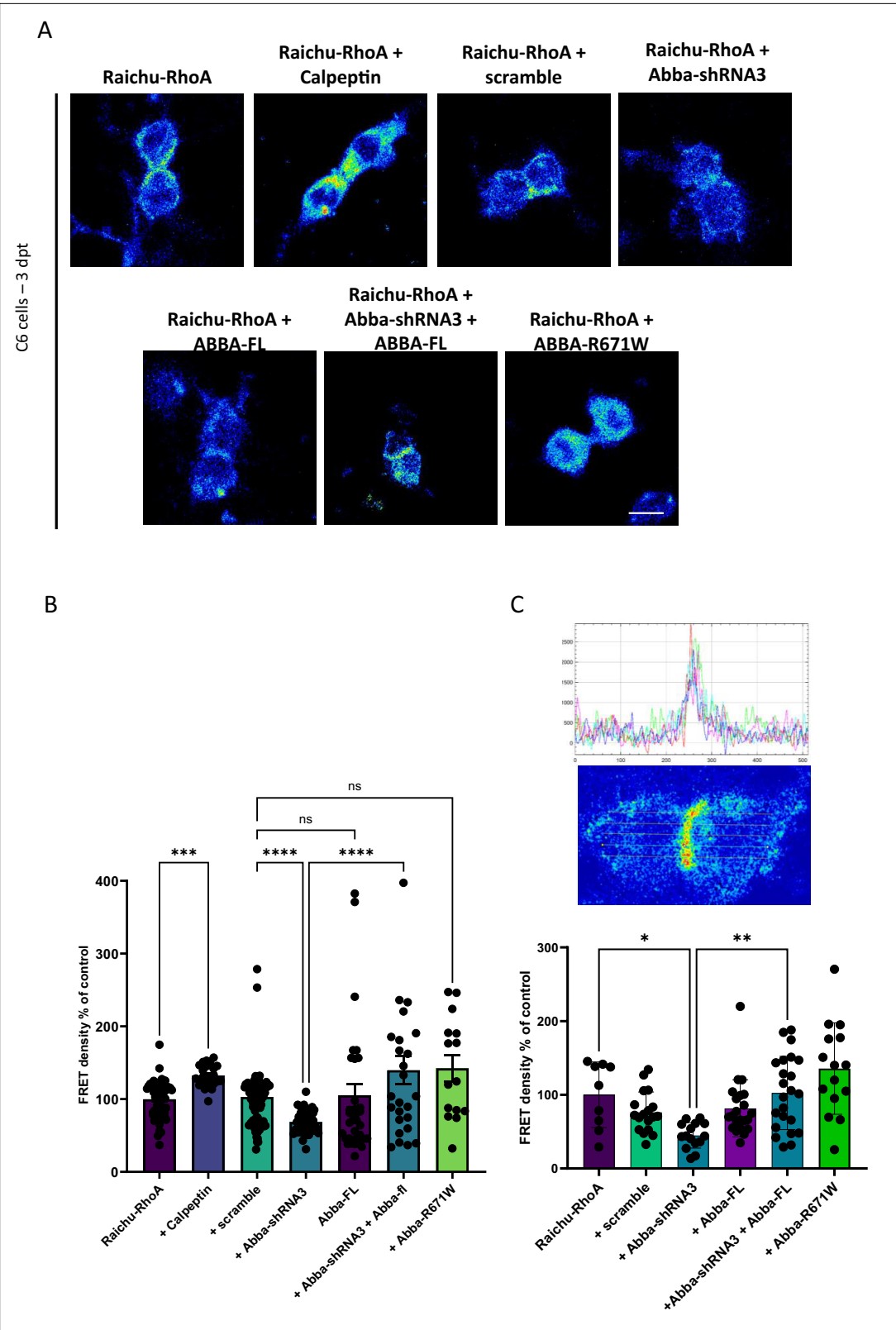

**Figure 5.** Mtss2 downregulation decreases RhoA activity. (**A**) Confocal images were captured of C6 cells 72 hr post-transfection with the Raichu-RhoA vector for monitoring RhoA activity. Six experimental groups were examined: transfected with scramble control, Mtss2-*shRNA*3, Mtss2-fl, rescue of Mtss2-*shRNA*3 with Mtss2-fl, and expression of the mutant Mtss2-R671W, in addition to testing the sensitivity of the assay to RhoA activation by calpeptin. (**B**) Quantification of FRET density (Raichu-RhoA activity) showed a decrease in Raichu-RhoA activity in C6 cells transfected with Mtss2-*shRNA*3

*Figure 5 continued on next page*

*Figure 5 continued*

as compared to transfected with the scramble (scramble, 103.1.03±5.3; n=102, Mtss2-*shRNA*3: 68.5±2.3; n=46; p<0.0001, Mtss2-*shRNA*+MTSS2-FL: 139.8±19.6 n=26, Mtss2-FL: 105.2±16.0; n=37, Mtss2-R671W: 142±17.9; n=15). (**C**) Quantification of the Raichu-RhoA activity in the furrow region shows similar effects. Upper panels show FRET image of a dividing cell and corresponding quantification of FRET intensity across the cell. Lower panel shows the normalized FRET intensity at the furrow (scramble, 76.3±27.6; n=18, Mtss2-*shRNA*3: 44.8±17.2; n=15, Mtss2-*shRNA*+MTSS2-FL: 102,5±49.7 n=23, Mtss2-FL: 81.6±38.7,0; n=22; p<0.0001, Mtss2-R671W: 135.7±62.4; n=15). Error bars represent mean ± s.d. \*\*p<0.002, \*\*\*p<0.001. Scale bar: 25 μm (**A**), 50 μm (**C**).

## Discussion

Mtss2 was initially discovered as a novel regulator of actin and plasma membrane dynamics, specifically in radial glial C6 cells. However, its role in brain RGP cells in live organisms has remained unclear. In this study, we observed that decreased expression of Mtss2 in radial glia within the cortical VZ leads to impaired radial migration and disrupted organization of radial glia. Notably, knockdown of Mtss2 specifically hindered cytokinesis in RGP cells, resulting in abnormal cell morphology, mitosis, and reduced numbers of glial and neuronal cells. The underlying mechanism behind this effect appears to involve impaired signaling through the actin regulatory protein RhoA, likely through its interaction with Nedd9 during the early and late stages of telophase. In accordance with these findings, we identified a missense variant in a patient with microcephaly and intellectual disability exhibiting similar clinical characteristics. Furthermore, overexpression of the mutant protein in cortical progenitors of mice resulted in phenotypes resembling, although not identical to, the downregulation of *Mtss2* in the cortex. The results presented in this study highlight the critical importance of *Mtss2* in cortical development and contribute to elucidating the potential mechanism underlying abnormal brain development associated with human variants of *MTSS2*.

### Radial glial exiting from cytokinesis requires Mtss2

BAR proteins can interact with PI(4,5)P$_2$-rich membranes through their BAR domain, and several studies indicate that PI(4,5)P$_2$ plays a central role in cytokinesis. Indeed, it has been reported that in mammalian cells, PI(4,5)P$_2$ accumulates at the cleavage furrow and recruits membrane proteins required for stability of the furrow because of a role in adhesion between the contractile ring and the plasma membrane. Moreover, overexpression of proteins that bind to PI(4,5)P$_2$ perturbs cytokinesis completion by interfering with adhesion of the plasma membrane to the contractile ring at the furrow, but not ingression of the cleavage furrow (*Normand and King, 2010*; *Zhang and Wu, 2015*). Immunocytochemistry of Mtss2 shows a strong accumulation at the cleavage furrow. In addition, in the absence of Mtss2, we observed an accumulation of cells blocked in cytokinesis (*Figure 3*). These observations and previous findings showing that Mtss2 can interact with PI(4,5)P$_2$-rich membranes through its BAR domain, as well as with the actin cytoskeleton through WH2, lead us to suspect that Mtss2 could link the actin contractile ring to the plasma membrane and participate in the mechanism for the invagination of the nuclear membrane during cytokinesis.

### Mtss2 interacts with Nedd9 to regulate RhoA activity

Here, we have found that Mtss2 interacts directly with Nedd9, and downregulation of Mtss2 leads to significant changes in RhoA activation. Nedd9 is a focal adhesion protein involved in mitotic entry and cleavage furrow ingression (*Hesse et al., 2012*). Similar to the results found in this study, previous results have shown that cells with abnormal expression of Nedd9 remain in G1, subsequently leading to cell apoptosis (*Zhong et al., 2012*). This is consistent with the view that Nedd9 triggers cells to enter mitosis. Nedd9 is known to be a positive regulator of RhoA signaling in mitosis. Indeed, activation of RhoA is required for cell rounding at early stages of mitosis and cleavage furrow ingression, and deactivation of RhoA is essential for cell abscission (*Dadke et al., 2006*). Moreover, it has been shown that Nedd9 overexpression blocks cells at abscission by maintaining levels of RhoA activation, and decreased Nedd9 expression inhibits cleavage furrow ingression by nonactivated RhoA pathway. In light of these results, we hypothesized that during cytokinesis at the cleavage furrow, the membrane composition changes and is enriched in PI(4,5)P$_2$, leading to Mtss2 recruitment and its interaction with Nedd9, which is essential to activate the RhoA pathway cleavage furrow ingression.

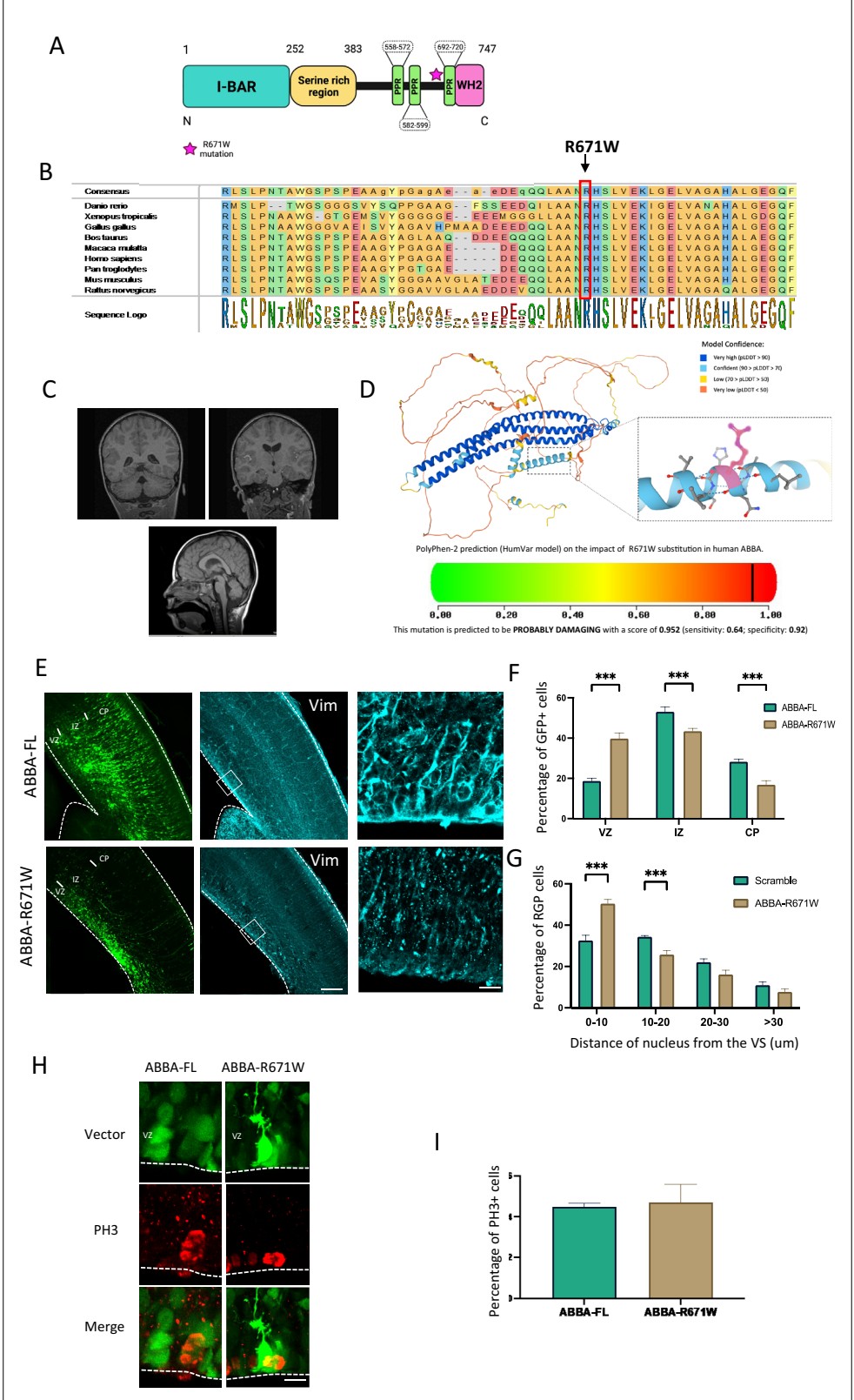

**Figure 6.** Effect of MTSS2 R671W human mutation on neuronal migration and mitosis. (**A**) MTSS2 is composed of an N-terminal Bin-Amphiphysin-Rvs (BAR) domain, a serine-rich region, three proline-rich motifs, and a C-terminal Wasp Homology-2 domain (WH2) domain. (**B**) Evolutionary conservation analysis revealed that the Arg671 site is conserved from zebrafish to humans. (**C**) Representative brain imaging features of one patient carrying MTSS2

*Figure 6 continued on next page*

*Figure 6 continued*

R671W variant. (**D**) 3D structure of MTSS2 denoting the position of arginine 571. Lower panel shows the high probability of disruption of the α-helix conformation by an arginine to tryptophan mutation. (**E**) Coronal sections of E17 mice brains electroporated at E14 with cDNAs encoding the human wild-type, MTSS2-FL, or the human mutant form of MTSS2, MTSS2-R671W. Expression of MTSS2-R671W results in a defect in neuronal migration as indicated by the accumulation of neurons in the SVZ/lower IZ, as well as a disorganization of the radial glial cell fibers using vimentin staining. (**F**) Quantification of cell distribution in the cortex at E17 (VZ/SVZ: MTSS2-FL 18.67 ± 1.44%, MTSS2-R671W 39.77 ± 2.73%; IZ: MTSS2-FL 53.07 ± 2.37%, MTSS2-R671W 43.40 ± 1.40%; CP: MTSS2-FL 28.25 ± 1.25%, MTSS2-R671W 16.8 ± 2.03%; n=6) showing a significant increase of mutant cells in SVZ/lower IZ. (**G**) Quantification of the distance of radial glial progenitor (RGP) nuclei from the ventricular surface (VS) at E17 (0–10: MTSS2-FL 32.56 ± 2.58%, MTSS2-R671W 50.04 ± 2.04%; 10–20: MTSS2-FL 34.37±0.59%, MTSS2-R671W 25.76 ± 1.97%; 20–30: MTSS2-FL 22.08 ± 1.59%, MTSS2-R671W 16.12 ± 2.12%; >30: MTSS2-FL 10.99 ± 1.52%, MTSS2-R671W 7.71 ± 1.38%; n=8 MTSS2-FL and n=6 MTSS2-R671W), revealing accumulation at this site for the mutant, but not wild-type MTSS2 (VZ/SVZ: p=0.000001: IZ: p=0.0014; CP: p=0.0003; 0–10: p=0.0002; 10–20: p=0.0005; 20–30: p=0.040; >30: p=0.150); mean ± s.d. (**H**) Brains were fixed at E17 and stained with the mitotic marker phospho-histone 3 (PH3). (**I**) The percentage of PH3-positive nuclei located at the ventricular surface (VS) did not differ in RGP cells expressing MTSS2-R671W (MTSS2-FL, 4.50 ± 0.17%, n=9; MTSS2-R671W, 4.71 ± 0.88%, n=5, p=0.761). Error bars represent mean ± s.d. **p<0.01, ***p<0.001. VZ: ventricular zone, SVZ: subventricular zone, IZ: intermediate zone, CP: cortical plate. Scale bars: 100 μm, 50 μm (**E**), 20 μm (**H**).

The online version of this article includes the following figure supplement(s) for figure 6:

**Figure supplement 1.** Impact of suppression and overexpression of MTSS2 on migrating neurons.

## Role of MTSS2 in neuronal migration and microcephaly

The proper formation of the cerebral cortex relies on the sequential generation of cortical layers, which is highly dependent on the normal production, survival, and migration of neuronal progenitors. Abnormalities in these processes are associated with pathological conditions such as microcephaly, leading to impaired cognitive development. Interestingly, both downregulation of *Mtss2* and overexpression of the *MTSS2* point mutation construct result in a similar accumulation of neuronal progenitors in the VZ and SVZ. This may be attributed to abnormal proliferation of neuronal progenitors caused by disrupted interaction with critical proteins involved in cell cycle progression. This disruption aligns with the hypothesized impact on Nedd9 interaction and reduced RhoA activity observed in conditions of low Mtss2 expression.

One possible explanation for these results could be the disruption of the predicted local tertiary structure. In this scenario, heterozygous expression of the human R671W variant would exert a dominant negative effect on *MTSS2*'s role in brain development, leading to microcephaly and cognitive delay. This notion is supported by recent work disclosing additional patients carrying the R671W variant (*Huang et al., 2022*). In the same study, the significant neurological phenotypes were observed in a *Drosophila* model where the ortholog of human MTSS2 and MTSS1, mim, was deleted. However, from a clinical genetics' standpoint, it is unlikely to find patients with the recurrent R671W mutation without any homozygous or compound heterozygous loss-of-function mutations elsewhere in the *MTSS2* gene. This could also suggest a gain-of-function effect of the R671W mutation. Supporting this notion, overexpressing MTSS2-R671W in cells expressing the wild-type *Mtss2 in this study* did not result in a dominant-negative decrease in RhoA activation, nor did it affect the expression of PH3 in vivo. These findings make it plausible to suggest that a mechanism responsible for the phenotype associated with overexpression of the human variant may primarily involve post-cell division processes, such as cell migration.

Although previous research has indicated that Mtss2 is not expressed in mature neurons (*Saarikangas et al., 2008*), more recent studies have shown transient expression of Mtss2 in neurons during conditions of increased neuronal plasticity (*Chatzi et al., 2019*). While our study primarily focused on Mtss2's role in radial glial cell proliferation, the data also suggest an additional role in the positioning of progenitor cells, although we cannot definitively attribute this to immature neuron migration. This is supported by the observed accumulation of electroporated progenitor cells in the VZ and SVZ, which may reflect defects in cell division or scaffold structure rather than active migration impairment. Additionally, we observed a significant decrease in process length in migrating cortical neurons with downregulated Mtss2 and expressing MTSS2-R671W mutant, but interestingly, dendritic processes were increased in length in neurons overexpressing *MTSS2* (*Figure 6—figure supplement 1*). Given that

disorganization of RGP cells does not lead to neuronal cell death, these findings suggest that Mtss2 may be involved in additional molecular mechanisms in differentiated neurons. Further experiments are needed to determine whether this phenotype is intrinsic to Mtss2's role in migrating neurons or if it is influenced by the disruption of guiding radial glia.

In conclusion, this study provides compelling evidence for the significant role of *Mtss2* in brain development. We demonstrate that Mtss2's involvement in the cell division of RGPs is a key component of this mechanism. Furthermore, these findings have translational relevance as they contribute to a better understanding of the putative mechanisms underlying human variants associated with microcephaly and developmental disabilities.

## Acknowledgements

We thank Drs. Lauren Briere and Gabrielle Lemire for the discussion about the data. We thank Biomedicum Imaging Unit (Helsinki, Finland) for their technical assistance with the time-lapse imaging of organotypic cultures. This project was supported by Eranet Neuron III program project ACROBAT, ANR project GABGANG, and Academy of Finland grant 341361 to CR, and by Fondation pour la Recherche Medicale 'aide au retour en France' to AC, and Academy of Finland (317038, 319907) and Sigrid Juselius foundation to JS and Finnish Cultural Foundation to EE.

## Additional information

### Funding

| Funder | Grant reference number | Author |
| --- | --- | --- |
| ERA-Net NEURON | ACROBAT | Claudio Rivera |
| Agence Nationale de la Recherche | GABGANG | Claudio Rivera |
| Research Council of Finland | 341361 | Claudio Rivera |
| Research Council of Finland | 317038 | Juha Saarikangas |
| Research Council of Finland | 319907 | Juha Saarikangas |
| Sigrid Juséliuksen Säätiö | | Juha Saarikangas |
| Fondation pour la Recherche Médicale | aide au retour en France | Aurelie Carabalona |
| Finnish Cultural Foundation | | Ellinoora Elomaa |

The funders had no role in study design, data collection and interpretation, or the decision to submit the work for publication.

### Author contributions

Aurelie Carabalona, Conceptualization, Data curation, Formal analysis, Investigation, Methodology, Writing – original draft, Writing – review and editing; Henna Kallo, Maryanne Gonzalez, Florence Molinari, Data curation, Formal analysis, Investigation, Methodology; Liliia Andriichuk, Investigation, Methodology; Ellinoora Elomaa, Formal analysis, Investigation, Methodology; Christiana Fragkou, Data curation, Formal analysis, Investigation; Pekka Lappalainen, Resources, Investigation; Marja W Wessels, Resources, Investigation, Writing – original draft; Juha Saarikangas, Data curation, Formal analysis, Investigation, Writing – review and editing; Claudio Rivera, Conceptualization, Resources, Data curation, Formal analysis, Supervision, Funding acquisition, Investigation, Project administration, Writing – review and editing

### Author ORCIDs

Aurelie Carabalona ⬤ https://orcid.org/0000-0003-2509-7198

Henna Kallo ![ORCID] https://orcid.org/0000-0002-5601-6920
Pekka Lappalainen ![ORCID] https://orcid.org/0000-0001-6227-0354
Juha Saarikangas ![ORCID] https://orcid.org/0000-0002-4665-2544
Claudio Rivera ![ORCID] https://orcid.org/0000-0003-2462-6561

### Ethics

Human subjects: Informed consent for publication and analysis of photos, imaging and clinical data was obtained from the patients' legal guardians. Brain magnetic resonance imaging (MRI) studies were performed on the subject and reviewed by the investigators in accordance with the Medical Ethics Committee of Erasmus Medical Center, Rotterdam, the Netherlands (Department Ethical permission number: MEC-2012387) to conduct human exome studies.

All experiments were conducted in accordance with the French and Finnish ethical committee which approved all procedures (No APAFiS#38335 and ESAVI/3183/2022 respectively). All animal experiments complied with the ARRIVE guidelines and were carried out in accordance with the U.K. Animals (Scientific Procedures) Act, 1986 and associated guidelines, EU Directive 2010/63/EU for animal experiments. All methods were performed following the relevant guidelines and regulations.

Joint Public Review: https://doi.org/10.7554/eLife.92748.4.sa1
Author response https://doi.org/10.7554/eLife.92748.4.sa2

---

## Additional files

### Supplementary files

Supplementary file 1. Yeast two-hybrid (Y2H) screening results.

Supplementary file 2. Summary of clinical and imaging phenotypes associated with mutations in MTSS2.

MDAR checklist

### Data availability

All data generated or analysed during this study are included in the manuscript and supporting files.

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
