## [Editor Report · eLife Assessment]

This **important** contribution to the field evaluated the function of the cytoskeletal protein ABBA in mediating key aspects of mitosis of neuronal precursor cells. The authors provide **compelling** evidence that ABBA interactions with its signaling partners is related to the development of at least some cases of microcephaly-a developmental anomaly associated with intellectual disability and other neurological findings.

---

## [Referee Report · Joint Public Review]

Carabalona and colleagues investigated the role of the membrane-deforming cytoskeletal regulator protein Abba (MTSS1L/MTSS2) in cortical development to better understand the mechanisms of abnormal neural stem cell mitosis. The authors used short hairpin RNA targeting Abba20 with a fluorescent reporter coupled with in utero electroporation of E14 mice to show changes to neural progenitors. They performed flow cytometry for in-depth cell cycle analysis of Abba-shRNA impact to neural progenitors and determined an accumulation in S phase. Using culture rat glioma cells and live imaging from cortical organotypic slides from mice in utero electroporated with Abba-shRNA, the authors found Abba played a prominent role in cytokinesis. They then used a yeast-two-hybrid screen to identify three high confidence interactors: Beta-Trcp2, Nedd9, and Otx2. They used immunoprecipitation experiments from E18 cortical tissue coupled with C6 cells to show Abba requirement for Nedd9 localization to the cleavage furrow/cytokinetic bridge. The authors performed an shRNA knockdown of Nedd9 by in utero electroporation of E14 mice and observed similar results as with the Abba-shRNA. They tested a human variant of Abba using in utero electroporation of cDNA and found disorganized radial glial fibers and misplaced, multipolar neurons, but lacked the impact of cell division seen in the shRNA-Abba model.

[Editors' note: the authors have responded to two sets of reviews, which can be found here, https://doi.org/10.7554/eLife.92748.2, and here, https://doi.org/10.7554/eLife.92748.1]

---

## [Author Response]

The following is the authors’ response to the previous reviews

**Public Reviews:**

**Reviewer #1 (Public review):**
The manuscript investigates the role of the membrane-deforming cytoskeletal regulator protein Abba in cortical development and its potential implications for microcephaly. It is a valuable contribution to the understanding of Abba's role in cortical development. The strengths and weaknesses identified in the manuscript are outlined below:Clinical Relevance:The authors identified a patient with microcephaly and intellectual disability patient harboring a mutation in the Abba variant (R671W), adding a clinically relevant dimension to the study.Mechanistic Insights:The study offers valuable mechanistic insights into the development of microcephaly by elucidating the role of Abba in radial glial cell proliferation, radial fiber organization, and the migration of neuronal progenitors. The identification of Abba's involvement in the cleavage furrow during cell division, along with its interaction with Nedd9 and positive influence on RhoA activity, adds depth to our understanding of the molecular processes governing cortical development.In Vivo Validation:The overexpression of mutant Abba protein (R671W), which results in phenotypic similarities to Abba knockdown effects, supports the significance of Abba in cortical development.Weaknesses:The findings in the study suggest that heterozygous expression of the R671W variant may exert a dominant-negative effect on ABBA's role, disrupting normal brain development and leading to microcephaly and cognitive delay. However, evidence also points to a possible gain-of-function effect, as the mutation does not decrease RhoA activity or PH3 expression in vivo. Additionally, the impact of ABBA depletion on cell fate is not fully addressed. While abnormal progenitor accumulation in the ventricular and subventricular zones is observed, the transition of progenitors to neuroblasts and their ability to support neuroblast migration remains unclear. Impaired cleavage furrow ingression and disrupted Nedd9 and RhoA signaling could lead to structural abnormalities in radial glial progenitors, affecting their scaffold function and neuroblast progression. The manuscript lacks an exploration of the loss or decrease in interaction between Abba and NEDD9 in the case of the pathogenic patient-derived mutation in Abba. Furthermore, addressing the changes in localization and ineraction in for NEDD9 following over-expression of the mutant are important to further mehcanistically characterizxe this interaction in future studies. These gaps suggest the need for further exploration of ABBA's role in progenitor cell fate and neuroblast migration to clarify its mechanistic contributions to cortical development.

(1) Response to statement on dominant-negative vs. gain-of-function effect of R671W variant:

We appreciate the reviewer’s thoughtful analysis of the potential mechanisms underlying the R671W variant. We agree that the heterozygous expression of the human R671W mutation may initially suggest a dominant-negative effect. However, our data indicate that this variant may instead exert a gain-of-function effect. As highlighted in the discussion section, overexpression of ABBA-R671W in cells that also express wild-type ABBA did not result in a dominant-negative decrease in RhoA activation nor affect PH3 expression in vivo. These findings suggest that the R671W mutation does not impair the canonical ABBA-mediated activation of RhoA, and instead, the resulting phenotype may involve post-mitotic processes, such as altered cell migration. This interpretation is further supported by previous clinical studies reporting additional patients with the same mutation and phenotypic outcomes.

(2) Response to statement on ABBA depletion and progenitor-to-neuroblast transition:

We agree that the question of how ABBA depletion affects cell fate and the progression of radial glial progenitors (RGPs) to neuroblasts is of significant importance. Our findings suggest that ABBA knockdown disrupts cleavage furrow ingression, which may block radial glial cells prior to abscission. This likely contributes to the observed accumulation of cells in the ventricular and subventricular zones, as seen in Figures 2A and 4D. Additionally, disrupted Nedd9 expression and impaired RhoA signaling appear to alter the structural integrity of RGPs, leading to detachment of apical and basal endfeet (Supplementary Figure 3). These structural abnormalities compromise the ability of RGPs to function as scaffolds for neuroblast migration. Although direct live imaging of neuroblast migration was beyond the scope of the current dataset, we believe our evidence strongly supports a model in which ABBA depletion disrupts progenitor structure and migration. Future studies will address these transitions more directly using live imaging and fate-mapping strategies

(3) Response to statement on loss of interaction between ABBA and NEDD9 with the R671W mutation:

We fully agree with the importance of investigating whether the R671W mutation alters ABBA’s interaction with NEDD9. While our study provides evidence for a role of NEDD9 in mediating ABBA function, we acknowledge that we did not directly test whether the R671W mutation disrupts this interaction. We apologize if our manuscript conveyed the impression that this point had been fully addressed. Due to technical limitations, particularly the poor performance of anti-NEDD9 antibodies in slice immunohistochemistry, we were unable to reliably assess the interaction or localization changes in vivo. Nevertheless, this remains a priority for future studies aimed at better understanding the mechanistic underpinnings of the R671W mutation.

(4) Response to statement on future directions for mechanistic characterization of NEDD9 localization and interaction:

We agree with the reviewer that further investigation into NEDD9 localization and its interaction with the ABBA R671W mutant is essential to better define the molecular consequences of this mutation. Unfortunately, as mentioned above, the current tools available to us did not permit reliable immunohistochemical detection of NEDD9 in tissue. We fully intend to pursue alternative approaches, such as tagging strategies or the use of more sensitive detection platforms, to determine whether the R671W mutation affects the subcellular localization or stability of the ABBA-NEDD9 interaction. These experiments will be critical to elucidate the pathway through which ABBA regulates progenitor cell behavior and cortical development.

**Reviewer #2 (Public review):**
Summary:Carabalona and colleagues investigated the role of the membrane-deforming cytoskeletal regulator protein Abba (MTSS1L/MTSS2) in cortical development to better understand the mechanisms of abnormal neural stem cell mitosis. The authors used short hairpin RNA targeting Abba20 with a fluorescent reporter coupled with in utero electroporation of E14 mice to show changes to neural progenitors. They performed flow cytometry for in-depth cell cycle analysis of Abba-shRNA impact to neural progenitors and determined an accumulation in S phase. Using culture rat glioma cells and live imaging from cortical organotypic slides from mice in utero electroporated with Abba-shRNA, the authors found Abba played a prominent role in cytokinesis. They then used a yeast-two-hybrid screen to identify three high confidence interactors: Beta-Trcp2, Nedd9, and Otx2. They used immunoprecipitation experiments from E18 cortical tissue coupled with C6 cells to show Abba requirement for Nedd9 localization to the cleavage furrow/cytokinetic bridge. The authors performed an shRNA knockdown of Nedd9 by in utero electroporation of E14 mice and observed similar results as with the Abba-shRNA. They tested a human variant of Abba using in utero electroporation of cDNA and found disorganized radial glial fibers and misplaced, multipolar neurons, but lacked the impact of cell division seen in the shRNA-Abba model.Strengths:Fundamental question in biology about the mechanics of neural stem cell division.Directly connecting effects in Abba protein to downstream regulation of RhoA via Nedd9.Incorporation of human mutation in ABBA gene.Use of novel technologies in neurodevelopment and imaging.Weaknesses:Unexplored components of the pathway (such as what neurogenic populations are impacted by Abba mutation) and unleveraged aspects of their data (such as the live imaging) limit the scope of their findings and left significant questions about the effect of ABBA on radial glia development.(1) Claim of disorganized radial glial fibers lacks quantifications.- On page 11, the authors claim that knockdown of Abba lead to changes in radial glial morphology observed with vimentin staining. Here they claim misoriented apical processes, detached end feet, and decreased number of RGP cells in the VZ. However, they no not provide quantification of process orientation to better support their first claim. Measurements of radial glia fiber morphology (directionality, length) and of angle of division would be metrics that can be applied to data. Some of these analysis could be done in their time-lapse microscopy images, such as to quantify the number of cell division during their period of analysis (though that is short-15 hours).

Response to: Lack of quantification of disorganized radial glial fibers and cell divisions in time-lapse data

We appreciate the reviewer’s insightful comment regarding the need for quantification of radial glial (RG) fiber morphology. In the revised manuscript, we have addressed this by providing new quantification of changes in vimentin staining, specifically measuring the dispersion of the signal as a proxy for fiber disorganization (see Supplementary Figure 1). These data support the observed morphological changes, including misoriented apical processes and detachment of endfeet, following Abba knockdown.

Regarding time-lapse analysis to track cell divisions, we attempted to follow individual cells during the 15-hour imaging window. However, due to the relatively short duration and limited number of cells that could be reliably tracked, the dataset did not allow for statistically meaningful conclusions. As an alternative approach, we performed live-cell imaging using Anillin-GFP, a reliable marker of mitotic progression. The distribution and accumulation of Anillin-GFP were analyzed in ABBA-shRNA3 and control conditions, and the results (now included in Supplementary Figure 3) indicate an increased number of cells arrested in late mitosis upon ABBA knockdown. This supports the notion of disrupted cytokinesis as a consequence of Abba depletion.

(2) Unclear where effect is:- In RG or neuroblasts? Is it in cell cleavage that results in accumulation of cells at VZ (as sometimes indicated by their data like in Fig 2A or 4D)? Interrogation of cell death (such as by cleaved caspase 3) would also help. Given their time lapse, can they identify what is happening to the RG fiber? The authors describe a change in "migration" but do not show evidence for this for either progenitor or neuroblast populations. Given they have nice time-lapse imaging data, could they visualize progenitor versus young neuron migration? Analysis of neuroblasts (such as with doublecortin expression in the tissue) would also help understand any issues in migration (of neurons v stem cells).- At cleaveage furrow? In abscission? There is high resolution data that highlights the cleavage furrow as the location of interest (fig 3A), however there is also data (fig 3B) to suggest Abba is expressed elsewhere as well and there is an overall soma decrease. More detail of the localization of Abba during the division process would be helpful-for example, could cleavage furrow proteins, such as Aurora B, co-localization (and potentially co-IP) help delineate subpopulations of Abba protein? Furthermore, the FRET imaging is unique way to connect their mutation with function-could they measure/quantify differences at furrow compared to rest of soma to further corroborate that Abba-associated RhoA effect was furrow-enriched?- The data highlights nicely that a furrow doesn't clearly form when ABBA expression and subsequent RhoA activity are decreased (in Fig 3 or 5A). Does this lead to cells that can't divide because of poor abscission, especially since "rounding" still occurs? Or abnormal progenitors (with loss of fiber or inability to support neuroblast migration)? Or abnormal progression of progenitors to neuroblasts?

Response to: Unclear location of the effect (RG vs. neuroblasts; cleavage furrow/abscission; migration issues)

We thank the reviewer for this comprehensive and thought-provoking set of questions.

a) Site of the effect – Radial Glia vs. Neuroblasts:

Our data suggest that the primary effect of ABBA depletion occurs in radial glial progenitors (RGPs), specifically prior to abscission. We observed accumulation of electroporated cells in the ventricular zone (VZ), which we interpret as a result of cytokinetic failure (e.g., Figure 2A, 4D). We also documented detachment of apical and basal endfeet (see Supplementary Figure 3), further supporting structural disruption of RG fibers.

b) Cell death analysis:

We considered using cleaved caspase-3 as a marker for apoptosis, but due to its transient and non-specific activation during development, we opted to assess overall survival via quantification of RGP cell numbers and localization. This approach better reflects the developmental impact of ABBA knockdown (Supplementary Figure 3).

c) Migration defects:

We agree that distinguishing between progenitor and neuroblast migration would be highly informative. Although we did not perform doublecortin or similar staining to differentiate these populations in this dataset, the accumulation of electroporated cells in VZ/SVZ strongly suggests a migration deficit. Addressing this in detail will require new experiments using lineage-specific markers and longer time-lapse recordings, which we plan to explore in future studies.

d) Cleavage furrow and abscission:

Our high-resolution imaging of Anillin-GFP and FRET-based RhoA activity shows that ABBA localizes predominantly at the cleavage furrow. New quantifications of RhoA activity (now in Figure 5) show that the reduction in signaling is most pronounced at the furrow in ABBA knockdown cells. These findings align with the hypothesis that ABBA, through Nedd9 and RhoA, is essential for proper furrow formation and abscission.

e) Mechanistic implications:

As the reviewer notes, ABBA knockdown leads to cells that "round" but do not complete division, likely due to poor cleavage furrow ingression. This could generate abnormal progenitors that are structurally compromised (detached fibers) and thus unable to support neuroblast migration or proper differentiation. The cumulative result is disrupted progression from RGPs to neuroblasts, impaired structural scaffolding, and possibly reduced cell viability.

(3) Limited to a singular time point of mouse cortical developmentOn page 13, the authors outline the results of their Y2H screen with the identification of three high confidence interactors. Notably, they used a E10.5-E12.5 mouse brain embryo library rather than one that includes E14, the age of their in utero electroporation mice. Many of the authors' claims focus on in utero electroporation of shRNA-Abba of E14 mice that are then evaluated at E16-18. Justification for the focus on this age range should be included to support that their findings can then be applied to all of mouse corticogenesis.

Response to: Use of E10.5–E12.5 library for yeast-two-hybrid (Y2H) screen

We appreciate the reviewer’s concern regarding the developmental stage of the Y2H library. We chose the E10.5–E12.5 brain embryo library based on prior work demonstrating that ABBA expression is strongest during early cortical development, particularly in radial glia at these stages (see Saarikangas et al., J Cell Sci 2008). The radial glia-specific expression of ABBA was previously validated using RC2 and Tuj1 markers at E12.5. Thus, the library we used is well-suited for identifying interactors relevant to radial glial function, including Nedd9. We have clarified this rationale in the revised manuscript.

(4) Detail of the effect of the human variant of the ABBA mutation in mouse is lacking.Their identification of the R671W mutation is interesting and the IUE model warrants more characterization, as they did with their original KD experiments.- Could they show that Abba protein levels are decreased (in either cell lines or electroporated tissue)?- While time-lapse morphology might not have been performed, more analysis on cell division phenotype (such as plane of division and radial glia morphology) would be helpful.

Response to: Lack of detail on R671W human variant effects

We thank the reviewer for encouraging further characterization of the R671W variant. In the revised manuscript, we now provide additional data on interkinetic nuclear migration (INM) defects resulting from R671W overexpression (see Supplementary Figure 3). These changes are consistent with disrupted radial glial organization and mirror aspects of the ABBA knockdown phenotype.

a) Protein levels:

We quantified ABBA expression in cells overexpressing the R671W variant (Supplementary Figure 5) and found no significant reduction compared to wild-type. This argues against a loss-of-function mechanism and supports a gain-of-function or dominant-interfering effect.

b) Morphological and division phenotyping:

While time-lapse imaging of R671W-expressing cells was not available in our dataset, we acknowledge that analyses such as division angle or radial glial morphology would be informative. Unfortunately, we were unable to perform these with the current data, but we agree these are important goals for future work.

Reviewer 2 conclusion:The resubmission has addressed many of the questions raised.I have a few comments that should be addressed:(1) The authors maintain a deficit in "migration of immature neurons" which remains unsubstantiated. In their resonse, they state: "we believe that the data showing the accumulation of migrating electroporated cells in the ventricular (V) and subventricular (SV) zones provide compelling evidence of abnormal migration in ABBA-shRNA electroporated cells. "- Firstly, they do not demonstrate that it's immature neurons, not RGs, that are affected. Secondly, accumulation of cells at the V-SVZ could be due to soley the inability for the RGC to undergo mitosis, therefore remaining stuck"The commentary of migration, especially of neurons, should be modified.

We appreciate the reviewer’s careful reading and valid concern regarding our use of the term "migration of immature neurons." We fully agree that the current dataset does not definitively distinguish whether the accumulated cells in the ventricular (V) and subventricular (SV) zones are immature neurons or radial glial progenitors (RGPs) arrested in mitosis.

To clarify, our observations indicate that electroporated cells accumulate in the VZ/SVZ following ABBA knockdown (Figures 2A and 4D), and this was interpreted as evidence of impaired migration. However, we now recognize that this accumulation may primarily reflect a block in cell cycle progression—specifically, at the stage of cleavage furrow ingression and abscission—rather than a migratory defect per se. This is supported by our new data using Anillin-GFP (Supplementary Figure 3), which show increased accumulation of cells with persistent Anillin expression, consistent with mitotic arrest. Furthermore, the detachment of apical and basal processes (also shown in Supplementary Figure 3) suggests that ABBA knockdown affects the structural integrity of RGPs, potentially compromising their scaffold function.

In light of these points, we have revised the manuscript to temper our conclusions regarding “migration defects.” Specifically, we now refer to the phenotype as “abnormal accumulation of progenitor cells” and clarify that, while these findings are consistent with impaired cell progression or scaffolding required for migration, we do not directly demonstrate impaired migration of immature neurons. As suggested, addressing this would require additional analyses, such as time-lapse imaging of post-mitotic cells or staining with markers like Doublecortin, which are beyond the scope of the current dataset but will be a focus of future investigations.

We thank the reviewer again for encouraging a more precise interpretation of our findings

**Recommendations for the authors:**

**Reviewer #1 (Recommendations for the authors):**
Supplementary Fig 4B - The figure doesn't show an increase in percentage of PH3 positive cells in the NEDD9-shRNA condition. The control images are also missing for comparison. The figure legend needs to be corrected to match with the figure showing no significant changes.

Thank you for this comment. This has been amended in the revised manuscript in the form of a new revised Supplementary Fig 4.

**Reviewer #2 (Recommendations for the authors):**
Minor annotations for slice culture assayThe authors should make note of ages of slice cultures in text and have better annotations of slice cultures (for example, in Fig 4-where is mitosis?)

We are sorry for the mistake it's not mitosis, it's the cleavage furrow stage. In addition, a new amended Figure 4 is provided.

The effects are hard to see in lower mag slice images in Fig. 6. Would recommend focusing on higher mag to highlight RG differences.

Thank you for this comment. This has been amended in the revised manuscript in the form of a new revised Figure 6.